structural engineering

structural analysis, prestress theory, structural dualities, plasticity, limit states, ductility

**Author for correspondence:**
Jaime Cervera Bravo
e-mail: jaime.cervera@upm.es

# Prestress behaviour and ductility requirements in structures: solutions from a unified algebraic approach

## Jaime Cervera Bravo and Laura Navas-Sánchez

Universidad Politécnica de Madrid. Escuela Técnica Superior de Arquitectura, Department of Physics and Building Structures, Avda. Juan de Herrera, 4, 28040 Madrid, Spain

 JCB, 0000-0002-1060-7397; LN-S, 0000-0002-3667-6358

This paper presents an algebraic approach that unifies both the elastic and limit theories of static structural analysis. This approach reveals several previously unpublished improvements, which are based on several dualities in the mathematical description. Firstly, we show a novel duality between the solutions to two different problems: the elastic solution to the internal forces of an externally loaded structure, and the nodal displacements induced by prestressing in one or several elements of the same structure. This duality is proven and discussed. The application of this solution to the limit state analysis is very productive, and includes the determination of the ductility requirements necessary to achieve full plastic behaviour, and the assessment of the prestress needed to limit or eliminate such requirements. The unified framework also allows to obtain the elasto-plastic deformed state at the beginning of the plastic structural collapse. We have also detailed the theoretical duality between two classes of structures—hyperstatic and hypostatic—which was derived from the linear algebra principles that define these solutions. Finally, we studied and exposed the dimensionality reduction of structural problems given by the singular value decomposition and the eigenvalues problem. An illustrative example that clearly illustrates all these points is provided.

## 1. Introduction

Different theoretical approaches to structural analysis allow engineers to model and interpret the behaviour of structures in different contexts and fields of application. Elastic models aim to characterize the behaviour of the initial load steps, while plastic models focus on the performance at the limit states. Elasto-plastic models not only cover a wider range of loading

conditions without the need of a full step-by-step—or incremental—analysis, but also are capable of approximating the dynamic plastic behaviour via pushover analysis. Nevertheless, complex elasto-plastic models pose several difficulties associated with the representations of the elastic or plastic kinematic variables, as well as the variability of these parameters in real situations.

This paper aims to present a unified algebraic review of static analysis in order to show, prove and use several dualities that exist within the structural theory. This methodology provides new insights into the established theories. The results presented in this paper include:

— An unpublished duality found between the solutions of two different elastic problems:
  (i) the determination of the internal forces of an externally loaded structure,
  (ii) the determination of nodal displacements induced by the prestressing of elements of the same structure.
— As a result of the latter solution, the comparison between the plastic and the elastic solutions and the determination of the ductility required to develop a full plastic solution becomes easy to obtain.
— The similarities between the theoretical properties of variable space states underlying those results are shown, deepening our understanding of the two classes of non-isostatic structural configurations:
  (i) hyperstatic structures and self-stress states,
  (ii) hypostatic structures and mechanisms states.
— An exploitation of the mathematical properties of such states allows engineers to perform the computation of the deformed state at the beginning of the structural collapse without resorting to step-by-step procedures.

Thus, this work is organized as follows: in §2, a summary of classical analysis theories is presented to introduce the concepts, the symbology and the established dualities between equilibrium and compatibility formulae.

In §3.2, we propose a new duality between the known elastic solution for the internal forces, represented by the solution matrix $S$ and the prestressing problem; the prestressing induced by predetermined displacements can be obtained using $S^T$, the transposed matrix of $S$. In the process of proving this relationship, the *self-stress* state spaces are deduced and subsequently analysed.

Consequently, these results can be used to compute the ductility requirements necessary to achieve a full plastic structural response.

In §4.1, the hyperstatic and hypostatic structural models are compared in terms of the properties of their *self-stress state* and *mechanism state*, respectively. Furthermore, we demonstrate the ability of eigenvalue and single value decomposition (SVD) problems to solve the elasto-plastic state reached by the structures at the beginning of a full plastic collapse when applied to the compatibility conditions.

Finally, we present several illustrative examples of the previous theoretical concepts and the conclusions (§§5 and 6).

# 2. The classical theories of elastic and limit analysis

In the literature, several mathematical approaches devoted to the representation, discretization and resolution of the analytical formulations of continuous problems can be found. Within this framework, they should be noted the developments proposed in a series of pioneering works ([1]: clear distinction between local and global rotations, [2]: trussed frame model for elastic plates, [3]: finite difference method in plates, [4]: displacement versus force methods, [5]: tensor algebra, [6]: stress fields in cells, …) as well as some later fundamental contributions such as [7]: truss equivalences for plate and shell cells, [8]: relaxation methods for solving systems of equations, [9]: influence coefficients in cell models for aircraft structures, [10]: energy principles applied to structural analysis, and others that are partially referenced in [11]. The latter contains a key bibliography that includes 77 works from 63 authors. Furthermore, a detailed description of such progress and its possible interpretations can be found in [12,13]. Within this variety, the approaches included in classical books such as [14] or [15] are particularly interesting as they present methods like the finite-elements method (FEM), the boundary elements method (BEM), the meshless local Petrov–Galerkin (MLPG) method, etc. under a unified framework. For this reason, the aforementioned discretizing approaches are summarized below.

When analysing the behaviour of a loaded structure we generally use models in which the geometrical and mechanical properties are represented by a finite number of regions $-e-$. Furthermore,

the internal forces $f$ of these regions (elements, bars, etc.) are commonly represented by their values in a number of selected points also known as the nodes. These regions also undergo internal deformations, which can be represented by the local movements $u$ of the nodes that are associated with them.

Thus, the $u$ and the $f$ vectors represent the behaviour of the regions and are related to the local coordinate systems associated with those regions. In addition, they are arranged in such a way that their scalar product $u^T \cdot f$ is a measure of the energy involved in the process. Specifically, in the case of elastic models, that scalar product's value is equivalent to double the elastic deformation energy, and is also the summation of internal complementary and deformation energies. The degrees of freedom of the regions are defined by the length of the vectors $u$ and $f$ and sum to $n = \sum_e n_e$, where $n_e$ is the number of degrees of freedom of each region $e$.

Furthermore, in the model, all regions are connected by their nodes forming a structural system with $N$ degrees of freedom. In addition, the global properties of the model can be condensed into the nodes: this is the case of external loads $F$ and global movements $U$, which are obtained by the resultant of forces and moments and the displacements and rotations at the nodes, respectively.

The coordinate system should be selected and ordered in a way such that the work done by the loads $F$ in a movement $U$ can be measured by the scalar product $U^T \cdot F$. It does not matter if the coordinate system is unique or has been set ad hoc for each node. Moreover, the list of $U$ does not include any restrained nodes (i.e. any point whose movement is impeded by any kind of reaction force). Therefore, to include them in the model, we have to introduce lists and matrices that distinguish between *restrained* and *free* nodes. In doing so, both sets of unknown elements are differentiated: the free displacements and the forces on the restrained nodes.

For the sake of simplicity, we assume null movements in the restrained degrees of freedom, then consider only the sublists or submatrices related to the free components of the kinematic model. Hence, the $F$ list only includes loads on free nodes and does not contain any reactions.

More specifically, we can state that $f \in \mathfrak{f}$, $F \in \mathfrak{F}$, $u \in \mathfrak{u}$, $U \in \mathfrak{U}$, where $\mathfrak{f} = \mathrm{space}(f)$, $\mathfrak{u} = \mathrm{space}(u)$, $\mathfrak{F} = \mathrm{space}(F)$, $\mathfrak{U} = \mathrm{space}(U)$, are the vector spaces of the vectors that represent the possible states of the structure.

The assembly is constrained by conditions that ensure both the equilibrium, EQU, between the loads $F$ and the internal forces $f$, and the compatibility, COM, between the global displacements $U$ and the deformations $u$. Thus, in a linear approximation we have the well-known equations for the free nodes as follows:

$$F = Hf \tag{2.1}$$

and

$$u = BU, \tag{2.2}$$

where $H$ and $B$ are the equilibrium and compatibility matrices, such that

$$H = B^T, \tag{2.3}$$

which can be easily proven by applying the virtual work principle. This results in the well-known duality between the mechanical and kinematic components of structural behaviour. In an equilibrated system $f, F$, the work done in any compatible movement $\bar{U}, \bar{u}$ must be the same for both the external $\bar{U}^T F$ and the internal $\bar{u}^T f$ representations of that work. Therefore, by introducing equations (2.2) and (2.1) in the form $\bar{u}^T f = \bar{U}^T F$, we obtain $\bar{U}^T B^T f = \bar{U}^T Hf$, which must be true for all $\bar{U}, f$ combinations, verifying equation (2.3).

More precisely, equilibrium $\mathcal{H} : \mathfrak{f} \to \mathfrak{F} : \{F = Hf; f \in \mathfrak{f}; F \in \mathfrak{F}\}$ and compatibility $\mathcal{B} : \mathfrak{U} \to \mathfrak{u} : \{u = BU; U \in \mathfrak{U}; u \in \mathfrak{u}\}$ are linear maps. The external work is the scalar product $\langle \cdot, \cdot \rangle : \mathfrak{U} \times \mathfrak{F} \to \mathfrak{R}$, and the internal one being the product $\langle \cdot, \cdot \rangle : \mathfrak{u} \times \mathfrak{f} \to \mathfrak{R}$.

It is worth mentioning that the relationships between the $n$ local and $N$ global degrees of freedom, which correspond to the dimensions of the $\mathfrak{f}$, $\mathfrak{u}$ and $\mathfrak{F}$, $\mathfrak{U}$ spaces, are limited to equations (2.1), (2.2). These two equations can be employed for any arbitrary reference system selected to represent the base of the $\mathfrak{F}$, $\mathfrak{U}$ spaces. It is also worth noting that the most important relationships when characterizing structural behaviour are those that relate the dimensions ($n$ and $N$) and the rank ($\mathcal{N}$) of the $B$ and the $H = B^T$ matrices. Although it is clear that $\mathcal{N} = rank(B) \leq n$ and $\mathcal{N} \leq N$, where $n \times N$ are the dimensions of the $B$ matrix, all the possibilities listed in table 1 are valid. This fact will be discussed in detail in §4.1.

**Table 1.** The relationships between rank $\mathcal{N}$ and dimensions $n \times N$ of compatibility matrix $\boldsymbol{B}$.

| relationships | structural properties |
|---|---|
| $\mathcal{N} < N \leq n$ | $n - \mathcal{N}$ self-stress states with $N - \mathcal{N}$ mechanisms |
| $\mathcal{N} = N < n$ | $n - \mathcal{N}$ self-stress states. Classic hyperstatic case. |
| $\mathcal{N} = N = n$ | no self-stress states nor mechanisms. Classic isostatic case. |
| $\mathcal{N} = n < N$ | $N - \mathcal{N}$ mechanisms. Classic hypostatic case |
| $\mathcal{N} < n \leq N$ | $N - \mathcal{N}$ mechanisms coexist with $n - \mathcal{N}$ self-stress states |

In table 1, the concept of *mechanisms* refers to free movements, which are movements that are unimpeded by deformations and forces in the structure. *Self-stress* states are considered to be states in which the internal forces (and strains) are not related to any external loads. Free movements, deformations induced by loads, and states of self-stress are all linearly independent. In this article, we have adopted the stress state approach because this can distinguish between alternative equilibrium solutions in *hyperstatic* conditions. However, other authors, as [16], make use of the *self-strain* approach, which is perhaps more difficult to interpret.

A vast majority of structural theories deal solely with the purely hyperstatic case because the set of equations required to obtain a solution for the $n$ unknown internal forces $f$ cannot be attained uniquely by the $N$ equilibrium equations. This means that when the $N$ free displacements $U$ are added as unknowns, $n$ additional equations are required to solve the problem. These equations are constructed through restrictions in the $n$ internal forces. These restrictions depend on the model selected to establish the material behaviour, and there are two alternatives described for the classic models. In the case of elastic analysis, these forces are linearly related with the $n$ internal movements (or 'deformations') $u$, obtained from equation (2.2). In the case of plastic analysis, these forces are restricted by member resistances, assuming unlimited ductile behaviour for the movement $(U, u)$ at the limit state.

The aforementioned set of equations is named here as material equations (MAT), since these equations depend on the material model.

## 2.1. The elastic solution

The constitutive or material equations for the linear model represent the elastic properties of each piece or region (element) $e$ of the structure as $f_e = k_e u_e$, equations that can be summarized for the whole structure in

$$f = ku, \tag{2.4}$$

where $k$ is a symmetric matrix, as can be deduced from Betti's Law.

The global elastic properties of the structure are assumed to be linear.

$$F = KU, \tag{2.5}$$

and if we consider the work done in any compatible movement $\bar{U}, \bar{u}$ for the said state, the following equation applies:

$$\bar{U}^T F = \bar{u}^T f = \bar{U}^T B^T ku = \bar{U}^T B^T kBU,$$

which must be true for all $\bar{U}$, meaning that

$$F = B^T kBU,$$

hence, with the construction of $K$ as

$$K = B^T kB. \tag{2.6}$$

These relationships can be summarized by the following conceptual schema [17,18], which also represents the following restrictions: firstly, the restriction that the resistant capacities ($r$) selected ($d$) for

the elements impose to the interactions ($\psi$) of the internal forces $f$; secondly, the restrictions on the combinations ($\Phi$) of global displacements $U$ imposed by the prescribed ($P$) limits in the displacements $Y$.[1]

$$
\begin{array}{ccccccc}
u & \leftarrow & \text{COM}(B) & & \leftarrow & U \langle (\Phi, P) \; Y \\
& \searrow & \updownarrow & & \swarrow & \\
\text{MAT } k \downarrow & \text{Internal work} & = & \text{External work} & \downarrow K \; \text{GLOB\_MAT} \\
& \nearrow & \updownarrow & & \nwarrow & \\
r \rangle \; (d, \psi) \; f & \rightarrow & \text{EQU}(H = B^T) & & \rightarrow & F
\end{array}
\tag{2.7}
$$

These relationships can be applied to two kinds of problems:

— Analysis problems: where the geometry, loads and dimensions are known ($B$, $F$, $k$) and designers search for solutions to the displacements, the deformations and the internal forces ($U$, $u$, $f$)
— Design problems: where the geometry and loads are given ($B$, $F$) and designers search for the appropriate dimensions for the structural elements ($r$, $k$) that give as a result consistent and acceptable displacements, deformations and internal forces ($U$, $u$, $f$).

The solution for the former type of problems, also known as *displacement problems*, equation (2.5), can be solved with

$$
U = K^{-1}F,
\tag{2.8}
$$

and internal forces deduced

$$
f = ku = kBU = kBK^{-1}F
$$

such that

$$
f = SF
\tag{2.9}
$$

with

$$
S = kBK^{-1} = kB(B^T kB)^{-1}.
\tag{2.10}
$$

Therefore, the matrix $S$ summarizes the mechanical components of the solution for any externally loaded structure. As for equations (2.1, 2.3 and 2.9), it can be shown that

$$
F = B^T f = B^T SF
$$
$$
f = SF = SB^T f.
$$

These equations suggest that $S$ is a generalized inverse of $B^T$. Nevertheless, this is not a true pseudoinverse as it is not unique as depends on $k$. This point will be further explored in §4.3.

The solution for the second (*design*) problem can be approached as follows:

— First, an equilibrium solution must be freely chosen by selecting a set of internal forces $f$, such that $F = B^T f$.
— Next, a compatible deformation-displacement solution must also be (almost) freely chosen, drawn, or designed, so that $u = BU$.

---

[1]The symbolic conventions employed in the schema (2.7) are defined in the following equivalences:

$$
x \langle (f, g) y \equiv f \cdot x \leq g \cdot y;
\tag{f1}
$$

$$
x \rightarrow \text{TAG}(y) \rightarrow z \equiv \text{TAG}(y) \overset{x}{\underset{z}{\updownarrow}} \equiv y \cdot x = z, \;\; (\text{TAG});
\tag{f2}
$$

$$
\begin{array}{c}
x \\
\searrow \\
\quad z \equiv x^T \cdot y = z; \\
\nearrow \\
y
\end{array}
\tag{f3}
$$

$$
\begin{array}{c}
\text{COM}(x) \\
\updownarrow \\
w = u \\
\updownarrow \\
\text{EQU}(y),
\end{array}
\equiv
\begin{cases}
\text{COM \& } w = u \Rightarrow \text{EQU} \\
\text{EQU \& } w = u \Rightarrow \text{COM} \\
\text{COM \& EQU} \Rightarrow w = u
\end{cases}
\tag{f4}
$$

where (f1) describes the restrictions, (f2) describes the linear transformation of a vector through one linear map, the direction of the transformation being shown by the arrows, (f3) represents the construction of a scalar product and (f4) describes the weak forms for the equilibrium or compatibility conditions.

— Furthermore, the $f$ and $u$ sets must be coherent: specifically, $sgn(f_i) = sgn(u_i)$, $\forall i$, although depending on the model, other conditions may apply.
— Finally, dimensions $r, k$ are selected such that restrictions to the strengths (introduced later as $\psi f \leq dr$) and the material equations $f = ku$ are respected.

In this way, the full set of equilibrium, compatibility and elastic material equations are satisfied.

## 2.2. The plastic or limit analysis

In this model, the ductility of the elements is assumed to be unlimited when the maximum threshold of resistance is reached at any point or section. Hence, the representation of material behaviour is introduced by limiting the internal forces that can be achieved

$$\psi f \leq dr, \tag{2.11}$$

where each row of $\psi$ represents the interaction of the internal forces $f$ under each limit condition and the same row in $dr$ represents the corresponding limit that derives from the set of nominal resistances $r$; e.g. in the resistance checks for the interaction of the moment and the axial force in a bar, $\psi$ represents the coefficients of the interaction in each check, and $dr$ the corresponding limit, selected from the set of resistances of the project. Where $d$ is thus the *influence* matrix that applies the appropriate nominal resistances to the limit for the selected equation. Thus, in each structural region $e$, $\psi_e f_e \leq d_e r$ represents the limit or yielding surface that establishes the permissible states on internal forces. Equation (2.11) is the aggregate of the limits for all members in the structure.

The kinematics of collapse can be represented by internal deformation velocities $\dot{u} = \varphi \lambda$, where $\varphi$ rows represent each possible independent component of a collapse deformation vector and $\lambda$ are the positive multipliers for any of the aforementioned local collapse mechanisms. This value will be zero when the yielding condition is not reached, and positive in any situation that provokes the occurrence of the aforementioned mechanism: i.e. when the yielding surface is achieved by the corresponding internal forces, where $\psi_{e,p} f_e = d_{e,p} r$ for that element $e$ and that mechanism $p$. Hence, the collapse condition can be described by

$$(\psi f - dr)^T \lambda = 0. \tag{2.12}$$

The flow rule in associated plasticity, also known as von Mises rule, states that the deformation velocities must be orthogonal to the yielding surface, such that $\varphi = \psi^T$, since rows of $\psi$ represent vectors orthogonal to the plane corresponding to that yielding condition. Therefore,

$$\dot{u} = \psi^T \lambda. \tag{2.13}$$

Hence, both the load factor that brings the structure to a limit state and the global collapse mechanism for a supposed initial loading condition $F_I$ can be obtained by

— The static approach deduced from the static theorem in associated plasticity: *If for a given structure and load we can find an equilibrated internal force distribution, with enough strength in all points to resist such internal forces, that load is a lower limit of the loads that can be resisted by the structure.* Therefore, this approach searches for the internal forces $f$ that maximize $\gamma$ for the collapse load $F = \gamma F_I$ while fulfilling the equilibrium conditions, equation (2.1) and the restrictions in terms of resistances, equation (2.11)
— Conversely, the kinematic approach searches for displacement velocities $\dot{U}$ and multipliers $\lambda$ that minimize $\Gamma(\lambda) = r^T d^T \lambda$ for $F = \Gamma F_I$, which satisfy the compatibility, equation (2.2), the collapse condition, equation (2.12), and the flow rule, equation (2.13). The kinematic theorem of plasticity (*If for a given structure and load we can construct a compatible collapse mechanism where the energy loss by the loads is equal to the dissipated energy in the plastic flow, that load is an upper limit to loads that can be sustained by the structure*) thus implies $F_{COM}^T \dot{U} = \Gamma F_I^T \dot{U} = r^T d^T \lambda \geq F^T \dot{U}$, This problem can be solved with the normalization of displacement velocities via the equality $F_I^T \dot{U} = 1$.

A key observation is that both problems can be represented and solved in a more general form. Note that $F$ is the collapse load and can be represented by the addition of a constant component $F_0$ and a factored component $F_I$, where $F = F_0 + \gamma F_I$, etc.

Furthermore, both problems are linear programming problems that can be easily solved with the Simplex algorithm as shown in [19] and [17] via the following forms:

$$\left.\begin{array}{l} \max_{f,\gamma}(\gamma) \\[6pt] \boldsymbol{\psi}f \le d\boldsymbol{r} \\[6pt] -\boldsymbol{B}^T f + \gamma \boldsymbol{F}_I = -\boldsymbol{F}_0 \\[6pt] \gamma > 0 \end{array}\right\} \tag{2.14}$$

and

$$\left.\begin{array}{l} \min_{\dot{u},\boldsymbol{\lambda}} \Gamma(\boldsymbol{\lambda}) = \min_{\dot{u},\boldsymbol{\lambda}}(\boldsymbol{r}^T \boldsymbol{d}^T \boldsymbol{\lambda}) \\[6pt] \boldsymbol{F}_I^T \dot{\boldsymbol{U}} = 1 \\[6pt] \boldsymbol{B}\dot{\boldsymbol{U}} - \boldsymbol{\psi}^T \boldsymbol{\lambda} = \boldsymbol{0} \\[6pt] \boldsymbol{\lambda} \ge 0. \end{array}\right\} \tag{2.15}$$

In addition, this representation of the problems can be easily converted into a *standard form* or normal form. As example, we have included an instance in a format that can be solved using the R package, the software used throughout the whole article [20,21]. Note that in this package the decision variables must be positive. Thus, we must replace $f$ with $f_p, f_n$, such that $f = f_p - f_n$, and the matrix condition should be formulated as

$$\begin{array}{c} \max_{f_p,f_n,\gamma}\left([\boldsymbol{0}^T\,\boldsymbol{0}^T\,1]\cdot\begin{bmatrix} f_p \\ f_n \\ \gamma \end{bmatrix}\right) \\[12pt] \begin{bmatrix} \boldsymbol{\psi} & -\boldsymbol{\psi} & \boldsymbol{0} \\ -\boldsymbol{B}^T & \boldsymbol{B}^T & \boldsymbol{F}_I \end{bmatrix}\begin{bmatrix} f_p \\ f_n \\ \gamma \end{bmatrix}\begin{array}{c} \le \\ = \end{array}\begin{bmatrix} d\boldsymbol{r} \\ -\boldsymbol{F}_0 \end{bmatrix} \\[12pt] f_p \ge \boldsymbol{0};\quad f_n \ge \boldsymbol{0};\; \Gamma > 0. \end{array} \tag{2.16}$$

etc. As shown in the previous references, both problems, described by equations (2.14) and (2.15), are also dual problems.

In addition, the global resistance conditions and global collapse mechanisms can be represented by equations in the following forms:

$$\boldsymbol{\Psi}\boldsymbol{F} \le \boldsymbol{R} \tag{2.17}$$

and

$$\dot{\boldsymbol{U}} = \boldsymbol{\Psi}^T \Lambda, \tag{2.18}$$

where $\boldsymbol{R}$ is the total amount of plastic energy dissipated in a unit displacement for each of the global independent modes of collapse. This can also be associated with the positions in a vector $\Lambda$, where $\boldsymbol{\Psi}\boldsymbol{F}$ is the energy lost by the loads in such a mechanism. $\Lambda$ represents the multipliers associated with each global mode of collapse, and can be interpreted in the same manner as the local multipliers associated with $\boldsymbol{\lambda}$. We can assume that these local multipliers $\boldsymbol{\lambda}$ were obtained from the global collapse geometries referenced by $\Lambda$ indicators. If we then analyse the small (linear) displacement fields, we can write the following:

$$\boldsymbol{\lambda} = L\Lambda, \tag{2.19}$$

where $L$ is a linear operator that projects each $\Lambda$ component over the $\boldsymbol{\lambda}$ space. For example, if an independent collapse mode $i$ is represented by a unit in component $i$ of $\Lambda$, and all other components of that vector are zeros, and the representation is normalized to the unit energy loss during the movement of the collapse; then the $i$-th column of $L$ would contain the $\boldsymbol{\lambda}$ multipliers corresponding to such failure per unit energy loss. Furthermore, if we rewrite $d\boldsymbol{r} \rightarrow \boldsymbol{r}$ to freely establish any possible value in the resistance equations (2.11), then the energy dissipation rate $\dot{W}_p$ can be represented by

$$\dot{W}_p = \dot{u}^T f = \dot{\boldsymbol{U}}^T \boldsymbol{F}. \tag{2.20}$$

Taking into account the significance of $\boldsymbol{\lambda}$ and $\Lambda$ values and equations (2.11), (2.13) and (2.17), (2.18), we can state that

$$\dot{W}_p = \boldsymbol{\lambda}^T \boldsymbol{\psi}f = \Lambda^T \boldsymbol{\Psi}\boldsymbol{F} = \boldsymbol{\lambda}^T \boldsymbol{r} = \Lambda^T \boldsymbol{R}, \tag{2.21}$$

where using equation (2.19) allows us to deduce

$$\Lambda^T L^T r = \Lambda^T R, \ \ \forall \Lambda$$

which implies

$$R = L^T r. \tag{2.22}$$

Finally, for this representation, the following equivalence can be proven

$$L^T \psi \equiv \Psi B^T, \tag{2.23}$$

since we have simultaneously satisfied these equations

$$\dot{u} = B\dot{U} = B\,\Psi^T \Lambda$$

and

$$\dot{u} = \psi^T \lambda = \psi^T L\Lambda, \ \ \forall \Lambda.$$

Hence, the whole analytical model can be summarized as follows:

$$
\begin{array}{ccccc}
\dot{u} & \xleftarrow{\phantom{xx}} & \text{COM}(B) & \xleftarrow{\phantom{xx}} & \dot{U} \\
 & \nwarrow \psi^T & \psi^T L \equiv B\,\Psi^T & \Psi^T \nearrow & \\
\downarrow & \lambda \downarrow & \xleftarrow{} L \xleftarrow{} & \downarrow \Lambda & \downarrow \\
\text{Int. work} = \text{Int. dissip.} & & \geq & \text{Ext. dissip.} = \text{Ext. work} \\
\uparrow & r \uparrow & \longrightarrow L^T \longrightarrow & \uparrow R & \uparrow \\
 & \nearrow \psi & L^T \psi \equiv \Psi B^T & \Psi \nwarrow & \\
f & \longrightarrow & \text{EQU}(H = B^T) & \longrightarrow & F
\end{array}
\tag{2.24}
$$

This model is equivalent to the schema illustrated in equation (2.7) but describes plastic models rather than elastic models.

Hence, we can approach problems in a similar way:

— Analysis problems: where the geometry, loads and dimensions are known ($B$, $F_I$, $r$) and designers search for a load factor $\gamma$ and the corresponding collapse condition in terms of forces, and geometry $F = \gamma F_I$, $\dot{U}$, $\dot{u}$. These can be solved via (2.14) or (2.15) equations.
— Design problems: where the geometry and loads are given ($B$, $F$) and designers search for dimensions $r$ that result in safe internal forces $f$, which correspond to coherent configurations of collapse mechanisms $\dot{U}$, $\dot{u}$.

For the latter problem, a possible approach would be to represent the solution cost $C$ as a linear function of design resistances $r$ distributed in the model as defined by the $dr$ expression.

$$C = c^T r. \tag{2.25}$$

The problem can thus be formulated as the following:

$$
\left.
\begin{aligned}
&\min_{f,r}(c^T r) \\
&\psi f - dr \leq 0 \\
&B^T f = F \\
&r \geq 0.
\end{aligned}
\right\}
\tag{2.26}
$$

As we have shown previously, this can be solved via Simplex. This solution can be broken down into the following steps:

— the (almost free) selection of a set of internal forces $f$ in equilibrium with the loads applied: $F = B^T f$,
— the assignment of elements with strengths capable of resisting said forces $\psi f \leq dr$,
— if needed, an iteration to achieve a better $f$ set in order to reduce the overall cost of the structure.

It is worth mentioning that, although the kinematics of the solution is not required in this approach, it is implied and obtained indirectly as a consequence of the duality between both cases. This duality is represented as follows. Firstly, the problem enunciated in (2.26) can be represented as

$$
\begin{aligned}
&\min_{f,r}(c^T r) \\
&\begin{bmatrix} -\psi & d \\ B^T & 0 \end{bmatrix} \begin{bmatrix} f \\ r \end{bmatrix} \begin{matrix} \geq \\ = \end{matrix} \begin{bmatrix} 0 \\ F \end{bmatrix} \\
&r \geq 0
\end{aligned}
\tag{2.27}
$$

and herein its complementary problem

$$\max_{\bar{\lambda},\dot{U}}(F^T\dot{U})$$

$$\begin{bmatrix} -\psi^T & B \\ d^T & 0 \end{bmatrix}\begin{bmatrix} \bar{\lambda} \\ \dot{U} \end{bmatrix} = \begin{bmatrix} 0 \\ c \end{bmatrix}$$

$$\bar{\lambda} \geq 0. \tag{2.28}$$

In the latter problem, the first group of equations represent the combination of the flow rule and the compatibility equations $(-\psi^T\bar{\lambda} + B\dot{U} = 0)$; in other words, the conditions for the kinematics of collapse for the $\bar{\lambda}$ and $\dot{U}$ parameters. The second group of equations $(d^T\bar{\lambda} = c)$ can be deduced from the basic unitary cost relationships, where the maximization of $(0^T\bar{\lambda} + F^T\dot{U})$, i.e. $F^T\dot{U}$, represents the total cost of sustaining $F^T$ forces with $\dot{U}$ unitary costs. Well, as the values in $c$ represent the unitary costs for the resistances $r$ in the case of the primal problem, the displacement velocities $\dot{U}$ may be interpreted as the unitary costs for the loads $\dot{F}$ in the case of the complementary or dual problem.

We can compute these unitary costs in the following manner. Firstly, in the equation $c = d^T\bar{\lambda}$ we must interpret the costs $c$ as being parameters in a compatible virtual collapse mechanism. And secondly, we must relate these parameters $c$ to the term $\dot{U}$ by using the equations applicable to a kinematic upper bound model. Consequently, we can formulate those conditions such 'that the optimum design can be found if we can find a compatible virtual collapse mechanism $\bar{\lambda}$, $\dot{u}$, $\dot{U}$ where the unitary design costs $c$ follow the equation $c = d^T\bar{\lambda}'$. It is worth mentioning that this is a general condition for an optimal plastic design that corresponds to those derived by Prager [22]); it is presented in a general form in ([23], 37 ss.). It can also be seen that such a *virtual* collapse mechanism will be equivalent to the *real* collapse mechanism for a structure with external $F$ forces and the obtained resistances $r$. Resistances that are calculated explicitly in primal problems or implicitly in dual problems are the strict resistances needed in the active sections, denoted by the $\bar{\lambda}$ multipliers. This equality occurs because the set of relevant equations are the same in both problems.

In order to justify the expression $d^T\bar{\lambda} = c$ in (2.28), we can define the total costs of the solutions as a product of the unitary costs and the structural parameters, via expressions such as $C = c^T r = c_f^T f = c_F^T F$, where the costs $c$ are given and the parameters $c_f$ or $c_F$ can be deduced from the resistance and equilibrium conditions. We have

$$\bar{\lambda}^T(-\psi f + dr) = 0$$
$$F = Bf,$$

where the $\bar{\lambda}$ multipliers corresponding to the active resistance equations have positive values, while the rest have null values where the change of the internal forces $f$ does not affect the overall cost: this is because the strengths in such positions are sufficient and do not need to be modified. These multipliers can identify the internal forces whose $c_f$ unitary costs may influence cost. The resistance equation can thus be expanded to the form

$$\left.\begin{aligned} \bar{\lambda}^T\psi f &= \bar{\lambda}^T dr \\ (\psi^T\bar{\lambda})^T f &= (d^T\bar{\lambda})^T r. \end{aligned}\right\} \tag{2.29}$$

and

The latter equation can be approached in the context of $c_f^T f = c^T r$ by interpreting $c_f = \psi^T\bar{\lambda} = \dot{u}$ as a virtual deformation velocity field associated with the active $\bar{\lambda}$ parameters related to the collapse conditions (the active resistance equations).

Finally,

$$\left.\begin{aligned} c_F^T F &= c_F^T B^T f = c_f^T f \\ c_f &= Bc_F \end{aligned}\right\} \tag{2.30}$$

and

such that $c_f = \dot{u} = B\dot{U}$ where $\dot{U} = c_F$ is the virtual displacement velocity field found in the solution of the dual problem, and where the deformation field $\dot{u}$ is determined by the kinematic compatibility.

## 2.3. Dualities in classical theories

As shown in the previous sections, work principles are the basement of the classical theories and the dualities found between the equilibrium and the compatibility equations (2.1), (2.2) and (2.3) are derived from them; see arrow directions in schemata (2.7) and (2.24). In addition, they support the dual linear programming problems that can solve plastic problems in equations (2.14) and (2.15).

Furthermore, a new duality is revealed, as we are able to establish the following between the equations that correspond to the limit condition:

$$R = L^T \psi f \tag{2.31}$$

and

$$\dot{u} = \psi^T L \Lambda, \tag{2.32}$$

which comes from the fact that $\Lambda^T R$ and $\dot{u}^T f$ both represent the same energy dissipation, albeit from two different points of views: the global collapse and the local deformation, respectively.

## 2.4. Some interesting scalars

The relevance of the work principles described in the preceding sections deserves a brief digression. Firstly, the summation of the associated scalars corresponds to the loss of potential energy of loads between their positions in the unloaded structure and their positions in the deformed structure. The components of this summation are the deformation energy and the complementary energy. Therefore, this scalar can be written as

$$U^T F = u^T f = \int_\Omega \varepsilon^T \sigma \, d\Omega. \tag{2.33}$$

If $\varepsilon$ and $\sigma$ were constant across the entire body, then that scalar would be a multiple of the volume of the body. An example of this would be an isostatic truss where all bars are strictly designed and the buckling effect is neglected.

We can thus consider the three following scalars: volume, stress volume and compliance or energy volume:

$$\mathcal{V} = \int_\Omega d\Omega, \tag{2.34}$$

$$\mathcal{W} = \int_\Omega |\sigma| \, d\Omega \tag{2.35}$$

and

$$\mathcal{U} = \int_\Omega \varepsilon^T \sigma \, d\Omega. \tag{2.36}$$

We can rewrite these scalars as the following:

$$\mathcal{V} = \int_\Omega d\Omega, \tag{2.37}$$

$$\mathcal{W} = |\sigma_d| \int_\Omega \frac{|\sigma|}{|\sigma_d|} \, d\Omega \tag{2.38}$$

and

$$\mathcal{U} = \varepsilon_d^T \sigma_d \int_\Omega \frac{\varepsilon^T \sigma}{\varepsilon_d^T \sigma_d} \, d\Omega, \tag{2.39}$$

where $\sigma_d$ and $\varepsilon_d$ are the design references for stresses and deformations, and where $\sigma = \kappa \varepsilon$ and $\varepsilon^T \sigma = \sigma^T \kappa^{-1} \sigma$ in elastic models. Their relationships and properties are thus very interesting in optimization problems. The stress volume, also known as *structural work* or the *quantity of structure*, (see definition 5 in [24]) is the volume weighted by local stresses. The energy volume, which represents the loss in potential energy by the loads, is the weighted stress volume, where the weights are given by local unitary deformations; this can also be described as the body volume weighted by local energy densities. It is worth mentioning that the dimensionality of the three scalars differs: the first one is associated with the cube of a length and the second and third scalars are associated with work or energy. As one can easily see in structures formed by axially stressed bars, the stress volume is a good measure of the structural requirements, as it involves both components of the solution: lengths and forces. Moreover, there are a lot of research

insights that can be derived from this scalar, *load path* when it is presented in the form of equation (2.40), see [25,26].

$$\mathcal{W} = \int_{\Omega} |\boldsymbol{\sigma}| \, \mathrm{d}\Omega = \int_s \frac{|N|}{A} \, \mathrm{d}(As) = \int_s |N| \, \mathrm{d}(s). \qquad (2.40)$$

In the case of an elastic model, a proportional change in all sections (surfaces) by a common factor $\alpha$, without any variation in loads or forces, multiplies the first scalar (the volume) by a factor $\alpha$, maintains the second scalar (the stress volume) at a constant value, and reduces the third (the energy volume) by a factor $1/\alpha$. Thus, for any structure, a change in the conditions of the material will result in a variation in the volume or compliance scalars, but will maintain constant the stress volume if the loads and forces do not change. This suggest that this scalar, the stress volume, should be adopted as a reference in optimization studies, as Michell did in [27], at least in cases where the location of loads and supports does not change.

# 3. The problem of the displacements induced by prestressing: the closure of geometrical incompatibilities

Once the symbology and classical dualities of structural analysis theories have been presented and discussed, we can apply them to new situations. We will begin with a problem of a structure under prestressed conditions. In this section, we will explore the problem, its solution, and its application to the establishment of clear relationships between the elastic and plastic solutions that will allow for the determination of the ductility required for the development of full plastic behaviour.

## 3.1. Displacements induced by prestressing

We have selected the kinematic model from among the different proposals as we believe that it is the most useful way of representing the prestressed conditions.

To demonstrate our findings, we will study a model of an undeformed and unloaded structure in which all the nodes are in their nominal geometrical positions, without any movement or deformation.

The prestressing actions can be visualized as the following: the actively prestressed elements of the structure are initially out of position, representing an incompatibility with the nominal geometry of the structure, such that the element would have to be stressed to be relocated in that position. It is worth noting that a similar model for imperfections, such as temperature differences, was originally employed by Castigliano in 1879 as remarked by ([28], §§7.4, 8.1).

Furthermore, the stress forces a movement of the whole structural system that eliminates this incompatibility. For example, in the case of a cable tensor, the difference between the length of the cable in the nominal geometry of the unstressed structure and the length of the unstressed cable that is connected to the whole structural system. The incompatibility is the gap that must be closed by the stressing action to account for this connection. However, in this case, the gap is a contraction deformation between the nominal and the real length of the cable.

Thus, we can define the prestressing action in the structure through the set $u_i$ of incompatibilities that have been eliminated by the prestressing action before taking into account any external loading. Thus, the set $u_i$ is the difference between the nominal positions of the elements' nodes in the undeformed structure, and the incompatible positions of those nodes in the undeformed geometry of the element. It is important to emphasize that in the former case, the incompatibilities have not yet been solved by the prestressing action. The pseudo-deformations $u_i$ would replace the nominal positions of the nodes of the incompatible members, considered in isolation, to the undeformed real positions in an unstressed and unconnected region; here all regions are considered independently, and without any deformation or force in other regions of the structure.

Thus, the internal forces that would theoretically be required to impose the deformations required to reach the nominal positions can be easily obtained

$$f_i = -ku_i, \qquad (3.1)$$

because the opposite deformations to the incompatibilities are these which should be imposed on the real initial positions of the nodes in order to move them to their nominal positions. Moreover, an external set

of actions would constitute a condition for the existence of said internal forces.

$$F_i = B^T f_i = -B^T k u_i. \tag{3.2}$$

As in the previous example, we must bring the nominal and the real positions closer to eliminate the incompatibility. This incompatibility is a contraction deformation (negative) that would reduce the nominal length of the cable to its real length in an undeformed state. In other words, we have to impose a positive extension to the cable (a traction force); this is why the minus sign is used. This traction would impose to the rest of the structure a prestressing force with the sign (the direction) of the incompatibility.

However, these internal or external forces are not required to attain their full nominal intensities, because the prestressing process reduces the distances that formed the initial incompatibilities and, consequently, the required forces. The procedure is similar to that used in the analysis of frames, where the analysis proceeds decomposing the final state between a first phase of *perfect embedding*, without displacements, and a second one of subsequent relaxation, to reach the final state. To solve it, some external forces $F_i$ must be considered. The first set of forces do stress the selected elements to eliminate the $u_i$ incompatibilities, while the second group composed of the opposite forces, $-F_i$ acts over the entire structure in the overall prestressing process so that the superposition of both states solves the real prestressing state.

Therefore, we have to study the structural response to the $-F_i$ loads, which can be described as the *nominal* prestressing loads before analysing any losses due to the deformation of the structure. The displacements of the initial positions of the prestressed structure before the application of any external load are

$$U_0 = -K^{-1}F_i = K^{-1}B^T k u_i \tag{3.3}$$

and

$$U_0 = S^T u_i. \tag{3.4}$$

This last equation is derived from the application of the rules for the transposition of matrix products, equation (2.10), and takes into account the fact that both $k$ and $K^{-1}$ matrices are symmetrical.

The preceding equation demonstrates the solution to the structural problem of computing the initial displacement $U_{0,j}$ in the $j_{th}$ degree of freedom as a result of a prestressing displacement defined by a unitary initial incompatibility *closure* $u_{i,k}$ on the $k_{th}$ deformation component in the model, where the $S_{kj}$ term is the *influence* coefficient to apply. The problem can thus be solved by calculating the $k_{th}$ $f_k$ stress induced by a unique unitary $F_j$ load (the energy conjugate of the $U_j$ displacement), as both problems present the same $S_{kj}$ coefficient in the relationships between the variables involved.

This result implies that we have identified a previously undiscovered duality.

## 3.2. The duality between the internal forces problem and the displacements from prestressing problem

As the groups of equations (2.9), (2.10) and (3.3), (3.4) show, we can add a new duality to those shown in §2.3. This duality can be identified by the pair $S \leftrightarrow S^T$. This new duality can be considered to be merely an extension of the equilibrium-compatibility duality identified by the pair $H \leftrightarrow B$, where $H \equiv B^T$, as is reflected in the dual paths described by $F \to f$ and $u \to U$ in schema (2.7), due to the reversibility in the material equations (MAT). This is feasible since the stiffness matrices $k$ and $K$ can be inverted to obtain the flexibility matrices $k^{-1}$ and $K^{-1}$. However, this duality deserves particular attention because it is applicable under conditions where there is no initial compatibility (COM).

## 3.3. The unloaded prestressed state as a self-stressed state

In the unloaded state, the deformation and stresses in any region of the structure are the result of the superposition of the nominal deformation and the stresses required to eliminate the initial incompatibilities resulting from the nominal prestress loads:

$$u_0 = BU_0 - u_i, \tag{3.5}$$

$$f_0 = k u_0 \tag{3.6}$$

and $$F_0 = 0. \tag{3.7}$$

The result of the last equation ($F_0 = F_i - F_i$) is consistent with previous results equations (3.6), (3.5), (3.2, (3.3) and (3.1) successively

$$F_0 = B^T f_0 = B^T k(BU_0 - u_i) = B^T kBU_0 - B^T ku_i$$
$$= -KK^{-1}F_i + B^T f_i = -F_i + F_i = 0.$$

Hence, the effective internal and external prestressing forces are $f_0$ and $F_0$, with reduction of the nominal forces described by $f_i$ and $F_i$, where the differences in $f_i$ to $f_0$ are components of the so-called *losses*. It must be stressed that the prestressing action, modelled using the nominal prestressing forces and the subsequent losses, can be considered to be indicative of cases where the global rigidity of the prestressed structure against the closure of the gap is much greater than the rigidity of the prestressed member. However, in the most interesting cases it is useless, and in the most extreme case it makes no sense. For example, in the case of an isostatic truss that has a tensor member, the action over the tensor does not vary the forces at all, and instead moves the whole structure. The nominal prestressing forces and the losses are useless for the explanation of this behaviour, but the incompatibility gap and the subsequent movements can be easily and accurately obtained to describe the behaviour of the truss.

It is important to note here that, as $F_0 = B^T f_0 = 0$, the self-stress state $f_0$ is part of the kernel of the equilibrium transformation $\mathcal{H}$ such that the dimensionality of such possible self-stress states must be limited to the difference between $n$ and $\mathcal{N}$; i.e. the hyperstatic degree of the structure. This implies that there are only $n - \mathcal{N}$ independent incompatibilities for the prestressing action.

In an isostatic structure, where $n = N = \mathcal{N}$ the $u_i$ movements cannot stress the structure, but the derived $U_0$ are equally obtained with equation (3.4). In these cases, it can be shown that $S = H^{-1}$ is a full rank square matrix that must ensure $f = SF = SHf$ for all arbitrary internal $f$ states.

## 3.4. Ductility requirements for the development of full plastic behaviour in ductile structures

In §2, we studied the solution to the elastic and plastic approaches to analysing a structure. For the elastic approach, we found that the solution of internal forces for a given set of loads is

$$f_e = SF_e, \tag{2.9}$$

where the $e$ indices refer to the elastic solution.

If the structure is sufficiently ductile, it will have a plastic limit state with internal forces $f_p$ that support a limit load $F_p = \gamma F_e$. This plastic limit state can be found by solving the following linear program:

$$\left.\begin{array}{l} \max_{f,\gamma}(\gamma) \\[4pt] \psi f \leq dr \\[4pt] -B^T f + \gamma F_e = 0 \\[4pt] \gamma > 0. \end{array}\right\} \tag{2.14}$$

If the elastic solution is multiplied by the load factor $\gamma$, the result will exceed the limit state of one or more internal forces in several members. Therefore, in such members, some plastic deformation will occur; in addition, their forces will be limited to their maximum threshold: the yielding force. With a corresponding change in several other internal forces, such plastic deformation can be described as the generation of an internal incompatibility; the necessary gap developed as a plastic deformation (striction, plastic rotation, etc.). In this case, the difference between the plastic and the overstressed elastic solutions can be seen as the closure of such gaps in the transition from the overloaded elastic state to the plastic state, assuming the members have enough strength to reach that elastic state. Such gaps could be interpreted as *prestress deformations*, but they are actually the *ductile deformations* required to develop the full plastic behaviour.

Thus, the problem can be presented as the determination of the ductility deformations $u_d$ that are required to transition from the overloaded elastic state $\gamma(f_e, F_e)$ to the plastic state $(f_p, F_p)$, where the difference is $(f_p - \gamma f_e, 0)$ as $\gamma F_e = F_p$.

Since the ductility requirements $u_d$ can be described by the $u_i$ incompatibilities from the previous section, and equations (3.6), (3.5) and (3.4) can be applied to the $f_0 = f_p - \gamma f_e$ internal stresses, we can obtain

$$f_p - \gamma f_e = k(-I + BS^T)u_d = Du_d. \tag{3.8}$$

The $u_d$ deformations represent the gap between the initial nominal dimensions of the bars or elements in the undeformed structure and their real dimensions after reaching their permanent plastic deformations, where these real dimensions are measured in an unstressed state.

From equation (2.10), we can define the $D$ symmetric matrix as

$$D = k(-I + BK^{-1}B^T k), \tag{3.9}$$

where we have made use of the symmetries of both, $K$ and $k$. It can be seen that the matrix $D$ is negative since the positive component reduces the incompatibilities. In terms of energy, the energy required to force the local deformations to close the incompatibilities without deforming the remaining members of the structure is reduced in the transition from the overstressed elastic to the plastic states once the structure adapts to the self-stressed state.

Thus, the ductility (prestress) requirements $u_d$ are a solution to the equation that relates the internal forces $f_d$ induced by ductility deformations $u_d$

$$f_d = f_p - \gamma f_e = D u_d. \tag{3.10}$$

As forces $f_d$ represent a prestress or self-stressed state, i.e. $B^T f_d = 0$, they belong to the kernel of the equilibrium transformation $\mathcal{H}$. Therefore, $f_d$ is a member of the subspace solution to that homogeneous equation, where $u_d$ and equation (3.10) are the vector and the equation that can select such a member from the whole set.

As stated in §2, we are dealing with a hyperstatic problem; specifically, this is the case where $n > N = \mathcal{N}$, and where no internal mechanisms are exhibited. However, it does possess $r = n - N$ redundant internal forces that cannot be solved via the $N$ equilibrium equations $F = B^T f$, which is the set of equations for which the solution differs in $f_d$, depending on whether plastic or elastic material conditions are applied.

In this case the kernel of the equilibrium matrix $B^T$ is a subspace $f_r$ of the internal force distributions, and $f_d \in \mathfrak{f}_r \subset \mathfrak{f}$, where $F_r = B^T f_r = 0$. The solution of the homogeneous equation

$$B^T f_r = 0, \tag{3.11}$$

gives us this kernel.

The construction of a basis for such a vectorial subspace requires the definition of $r$ linearly independent distributions of the internal forces, $f_r$, where the above condition is met for each distribution. This is a procedure that can be routinely made with linear algebra packages.

As we are only interested in the plastic deformations and not in the internal forces, we will use the (3.10) and (3.19) equations to select the appropriate solutions. We solve the eigenproblem for $D$ and obtain a collection of $u_r$, $u_n$ vectors with corresponding $\Sigma_r$, $\Sigma_n$ values, where $\Sigma_n = 0$ and $u_n$ is the null base of $D$. This gives the following:

$$D = u_r \Sigma_r u_r^T, \tag{3.12}$$

where we restrict the $u_r$ base, and the corresponding $r \times r$ dimensions of $\Sigma_r$ to those referencing non-null eigenvalues in $\Sigma$; in doing so, dimensions of $u_r$ are $n$ rows $\times$ $r$ columns, with $r < n$.

We can thus write

$$f_d = u_r \Sigma_r u_r^T u_d \tag{3.13}$$

and

$$\Sigma_r^{-1} u_r^T f_d = u_r^T u_d, \tag{3.14}$$

where $\Sigma_r^{-1}$ is the diagonal matrix of the inverses of the non-null eigenvalues found, as in this representation we have $u_r^T \cdot u_r = I$ (an identity matrix of $r \times r$), where $u_r \cdot u_r^T \neq I$.

Since the $u_d$ displacement is included in the subspace spanned by the base vectors $u_r$, it can be represented as a linear combination of that base plus an arbitrary combination of the null base components of $D$, which we can represent as $u_n$. Furthermore, by using the coefficients $g$ and $\alpha$, the relationship could be written as

$$u_d = u_r g + u_n \alpha, \tag{3.15}$$

where

$$D u_n \alpha = 0, \ \forall \alpha. \tag{3.16}$$

Thus, we obtain the following result from equation (3.14) and the orthogonality of the deformation base

$$\Sigma_r^{-1} u_r^T f_d = u_r^T u_d = u_r^T u_r g = g. \tag{3.17}$$

From this result, we can now reconstruct the required ductile deformations.

To do so, we have to compute the $\boldsymbol{\alpha}$ factors (that amount to $n - r$ values) in equation (3.15) to ensure null values for $u_d$ in cases (member deformations) that do not exhibit plastic flow and non-null values in cases where this flow develops during the collapse mechanism. Specifically, the flow is not-null in all plastifying elements except for the final element, when the global plastic limit is reached: if there are $r$ hyperstatic redundant conditions, collapse happens with $r + 1$ plastic flow components in the deformations $\dot{u}$ obtained in the plastic kinematic approach as described in equations (2.15), (2.2), and (2.13). Hence, there are $n - r - 1$ null values in $\dot{u}$. Consequently, we already have $r$ known base coefficients in $g$ and $n - r$ unknowns in $\boldsymbol{\alpha}$, and only $n - r - 1$ additional conditions. If we can detect the last reached plastic limit, we can add it to the $n - r - 1$ indices of null values in $\dot{u}$; this completes the set of equations that can be used to solve the $\boldsymbol{\alpha}$ coefficients with these $n - r$ indices.

One method of detecting this last reached plastic limit involves using a measure of the local minimum dissipated energy, as the correct solution should correspond to maximum global dissipation.

For each of these $n - r$ indices, we have

$$(u_r g + u_n \boldsymbol{\alpha})_i = (u_d)_i = (\dot{u})_i = 0, \tag{3.18}$$

thus obtaining

$$\boldsymbol{\alpha} = -(u_n)_i^{-1}(u_r)_i g, \tag{3.19}$$

where $(\cdot)_i$ is the selection of the identified $i$ rows in $(\cdot)$.

Consequently, we can compare the ductility requirements with the ductility capabilities in the affected members, with the latter being detailed in structural codes, such as ASCE-41 or Eurocode 8 part 3.

The dimensions of several of those matrices are summarized here.

| matrix (rows, cols) | $u_d$ | $u_r$ | $g$ | $\Sigma_r$ | $f_d$ |
|---|---|---|---|---|---|
| | $(n, 1)$ | $(n, r)$ | $(r, 1)$ | $(r, r)$ | $(n, 1)$ |

Assuming that the ductility requirements $u_d$ are fulfilled, the plastic deformation energy required to reach this state can be easily obtained. This energy is the work done by the plastic solution to the internal forces $f_p$ in such deformations. It is worth noting that the members that remain in the elastic state have a null component in the plastic deformation vector. Therefore, these members will flow with the constant force obtained from the internal forces vector.

$$W_{\text{plast}} = u_d^T f_p. \tag{3.20}$$

Furthermore, we can easily obtain the deformed state at the point where the full plastic state is achieved. This state can be calculated as the sum of two states: the overstressed elastic state, considering the loads $\Gamma F_I$ and the displacements $\Gamma U = \Gamma K^{-1} F_I$, and the incompatible state, which is obtained by applying equation (3.4) to such ductile flows; this latter state arises because of the ductility flow.

The same displacement-deformation field can be found in an alternative way: by obtaining the total deformations as the sum of the ductility component $u_d$ and the elastic component $u_e$ from the internal forces $f_p$ (as if they have been attained elastically: $u_e = k^{-1} f_p$). Subsequently, the displacements $U$ can be constructed from the compatible deformations $u = u_e + u_d$ with an equation that will be presented and proven in §4.3: equation (4.11).

In closing, the analogy we have employed demonstrates the following: the plastic set of internal forces can be obtained without any plastic flow if we could substitute that plastic flow with a prestressing program that achieves the same gap set.

## 3.5. The integrated force method

In this section, we demonstrate that the integrated force method (IFM) from [29], can be seen as a parallel method to the self-stress state approach.

As it was stated in §2, the most common structural problem is purely hyperstatic, i.e. $n > N = \mathcal{N}$, with $r = n - N$ redundant internal forces that are not solved by the $N$ equilibrium equations, $F = B^T f$.

The kernel of the equilibrium $B^T$ matrix, is a subspace of the internal force distributions, and any of its components $f_r$ is the solution to the homogeneous equation,

$$B^T f_r = 0, \tag{3.21}$$

where $f_r$ is any self-stress state. To construct a basis for such a vectorial subspace, we have to find $r$ linearly independent distributions of the internal forces, where the above condition is met for each of such distributions. As mentioned previously, this procedure can be routinely made with linear algebra packages.

Thus, if each component of such a basis is assembled as a column in a matrix $n$ (that will have then $n$ rows and $r = n - N$ columns) the following conditions hold:

$$B^T n = 0 \tag{3.22}$$
$$n^T B = 0 \tag{3.23}$$
$$n^T B U = 0 \tag{3.24}$$

and

$$n^T u = 0. \tag{3.25}$$

The last equation shows that we can eliminate the global displacement unknowns $U$. Thus, this produces a set of $n = N + r$ equations to solve the internal forces problem. In an elastic scenario, these equations could be added to the $n$ MAT equations to obtain a complete set that can be formulated using only the unknown internal forces $f$, without the need for an intermediate solution nor to the displacements or the deformations

$$\left. \begin{array}{l} F = B^T f \\ n^T u = 0 \\ u = k^{-1} f \end{array} \right\} \tag{3.26}$$

and

$$\begin{bmatrix} B^T \\ n^T k^{-1} \end{bmatrix} f = \begin{bmatrix} F \\ 0 \end{bmatrix}. \tag{3.27}$$

It must be stressed that the differences between the displacements are never computed in this approach; this achieves a better performance in terms of round-off errors. In addition, the flexibility matrices $k^{-1}$ are used, which results in a better performance in nonlinear problems, where such matrices are usually better conditioned and easier to obtain than their inverses, the stiffness matrices $k$. However, the problems are now formulated in non-symmetrical formats and require the use of non-symmetrical solvers. Regardless, those properties are capable of producing better results than the classical stiffness matrix method.

# 4. Exploiting the vector space properties of the compatibility conditions

## 4.1. The classification of structures by their static properties

As suggested in §2, and used in §3, the equilibrium and compatibility equations can classify structures based on the relationships between the dimensions $\mathcal{N}$, $n$ and $N$ of the spaces spanned by the $\mathcal{B}$ and $\mathcal{H}$ transformations, i.e. the spaces spanned by the $u$ and $f$ local vectors, or the $U$ and $F$ global vectors, where these dimensions are the rank and the dimensions of the $B$ or $H$ matrices.

When $\mathcal{N} = N < n$, the structure is *hyperstatic*; this is the most common case. In this case, the structure has multiple equilibrium solutions that differ in the set of unloaded self-stress states. These solutions can be activated by an equal number of independent incompatibilities or closures, and can be obtained from the $n - N$ dimensional subspace that constitutes the kernel of the $\mathcal{H}$ equilibrium transformation from $\mathfrak{f} \to \mathfrak{F}$. The fundamentals of linear algebra indicate that $ker(\mathcal{H}) = \{f \in \mathfrak{f} | Hf = B^T f = 0\}$ is a subspace, with a dimension of $n - \mathcal{N}$, of the $n$ dimensional space $\mathfrak{f}$.

When $\mathcal{N} = N = n$, the structure is *isostatic*. This class of structures can be easily understood, and is fully solvable under the equilibrium conditions because the $B^T$ matrix is square and invertible.

When $\mathcal{N} = n < N$, the structure is *hypostatic*. In this class of structure, there are $N - \mathcal{N}$ load-equilibrated states that cannot be reached by any internal force or deformation state. Furthermore, the corresponding displacement modes can be activated with a null internal deformation work; this implies the presence of free moving components in the structure, i.e. mechanisms. Free movements of the mechanism can be anywhere within the kernel of the compatibility transformation $\mathcal{B}$, i.e. $U_m \in ker(\mathcal{B})$, with $ker(\mathcal{B}) = \{U \in \mathfrak{U} | BU = 0\}$.

In addition, in the hyperstatic case, the kernel of the equilibrium equation shows the existence of null external forces with non-null internal forces. In other words, the work states of non-null internal deformation can coexist with null external work, although this situation requires previous work (the prestress action) to accumulate such energy. The cokernel of such equations is the kernel of the compatibility equations. This suggests that there can be also non-null global work states that coexist with null internal work states. Thus, the unlimited movement of the structure is required to dissipate such energy. These key relationships are summarized in table 1.

## 4.2. The singular values decomposition of compatibility conditions

Here, we will use the *singular values decomposition* (SVD) to complete a standard task in linear algebra: the characterization of the properties of the matrices $B$ and $B^T$. In this context, the SVD is an excellent theoretical tool that provides easy and clear explanations. However, it is worth mentioning that a number of its applications can be achieved more efficiently with other simpler decompositions or methods.

The SVD of matrix $B$ has the form

$$B = v\Sigma V^T, \tag{4.1}$$

where

$$v^T v = I; \quad V^T V = I, \tag{4.2}$$

where $v$ and $V$ are the basis for the *left* and *right* spaces defined by the linear map of $B$, (the spaces for $u$ and $U$, respectively), and where $\Sigma$ has only non-null and decreasing elements in its diagonal. We consider the left $n$ and the right $N$ nullspaces for $B$, to be subspaces of those defined by $v$, $V$ with the following properties:

$$\begin{aligned} n \subset v; & \quad n^T B = n^T v\Sigma V^T = 0 \\ N \subset V; & \quad BN = v\Sigma V^T N = 0. \end{aligned} \tag{4.3}$$

and

A right null space means that it is *not influencing* or *non-significant*, as this space's components only produce null elements via the linear map. By contrast, a left null space is *unreachable*, as it defines a set of elements on the left space that cannot be reached via the linear map.

Thus, by considering both the equilibrium and the compatibility equations

$$F = B^T f = V\Sigma^T v^T f \tag{4.4}$$

and

$$u = BU = v\Sigma V^T U \tag{4.5}$$

the left $n$ and right $N$ subspaces and null spaces of the matrix $B$ have the following effects:

— The subspace $n$ spans all hyperstatic situations and
  (i) in the equilibrium equations (for its interpretation as $f$), it implies the presence of sets of internal forces that do not equilibrate any external forces, i.e. its space spans only self-stress states,
  (ii) in the compatibility equations (for its interpretation as $u$), it implies the presence of sets of internal deformations that are unreachable for any global displacements (incompatible states).
— The subspace $N$ spans all situations with the existence of mechanisms and
  (i) in the equilibrium equations (for its interpretation as $F$), it implies the presence of unreachable external forces, i.e. sets of external forces that cannot be equilibrated with any combination of internal ones,
  (ii) in the compatibility equations (for its interpretation as $U$), it implies the presence of sets of movements that do not produce deformation in any member.

## 4.3. The pseudoinverse of the compatibility conditions: constructing displacements from deformations

Once we have shown the SVD of the compatibility matrix $B$, equation (4.1), the pseudoinverse of that matrix can be constructed with

$$B^{-T} = V\Sigma^{-T} v^T, \tag{4.6}$$

where $\Sigma^{-T}$ is the transpose matrix of $\Sigma$ where non-null elements in the diagonal have been substituted by their inverses ($\Sigma_{ii} \rightarrow 1/\Sigma_{ii}$). Thus we can show that

$$\boldsymbol{BB}^{-T} = \boldsymbol{v}\Sigma\boldsymbol{V}^T\boldsymbol{V}\Sigma^{-T}\boldsymbol{v}^T = \boldsymbol{v}\Sigma\Sigma^{-T}\boldsymbol{v}^T \qquad (4.7)$$

and

$$\boldsymbol{B}^{-T}\boldsymbol{B} = \boldsymbol{V}\Sigma^{-T}\boldsymbol{v}^T\boldsymbol{v}\Sigma\boldsymbol{V}^T = \boldsymbol{V}\Sigma^{-T}\Sigma\boldsymbol{V}^T. \qquad (4.8)$$

We can subsequently rename them as

$$\boldsymbol{I}^{+l} = \Sigma\Sigma^{-T}; \quad \boldsymbol{I}^{+r} = \Sigma^{-T}\Sigma, \qquad (4.9)$$

which results in square matrices with the dimensions of the left and right spaces respectively ($n$ and $N$), with unitary elements only in the first diagonal positions, in a number corresponding to the rank of the $\boldsymbol{B}$ matrix.

In the usual hyperstatic case, $n = \dim(\boldsymbol{u}) > \text{rank}(\boldsymbol{B}) = \mathcal{N} = \dim(\boldsymbol{U}) = N$, and $\boldsymbol{I}^{+r} = \boldsymbol{I}$ but $\boldsymbol{I}^{+l}$ has only $\mathcal{N}$ unitary elements in its diagonal.

Thus, equation (4.8) can be reduced to

$$\boldsymbol{B}^{-T}\boldsymbol{B} = \boldsymbol{I}. \qquad (4.10)$$

However, because the $\boldsymbol{I}^{+l}$ matrix has some null rows in equation (4.7), it eliminates some elements in the product of basis $\boldsymbol{v}$ and $\boldsymbol{v}^T$, eliminating the possibility of providing a general assertion about the possible result.

Consequently, we can only employ this pseudoinverse in the compatibility equation

$$\boldsymbol{u} = \boldsymbol{BU} \Rightarrow \boldsymbol{B}^{-T}\boldsymbol{u} = \boldsymbol{U}. \qquad (4.11)$$

From equation (4.11), we can calculate the displacements related to any prescribed compatible deformation state. This gives us the possibility of reconstructing a displacement field from any known compatible deformation field, as the superposition of the elastic and ductile fields that result in the elasto-plastic deformation state reached at the beginning of the plastic collapse.

The SVD constitutes a generalization of the approach of eigenvalue problems from symmetrical matrices to non-symmetrical matrices. As such, it can introduce ideas from symmetrical problems to the resolution of non-symmetrical problems. One of the key ideas that it introduces is the reduction in the dimensionality of the problem's space. Furthermore, it is well known that elastic dynamics or stability problems can be presented as eigenvalue problems where dimensionality reduction is one of the most common approaches. Here, we will present one other important eigenvalue problem before considering this approach.

## 4.4. The load mode eigenvalue problem

In a general design problem, the loads $\boldsymbol{F}$ can be represented by the superposition of several patterns $\mathcal{F}$ in the form $\boldsymbol{F} = \mathcal{F}\boldsymbol{\lambda}$. In this situation, the displacements corresponding to each pattern could be represented by $\mathcal{U}$ where each column of $\mathcal{F}$ has a stiffness relationship with its corresponding displacements. We can thus write $\mathcal{F} = K\mathcal{U}$. Therefore, we have $\mathcal{F}$ and $\mathcal{U}$ as sets of columns representing the possible load patterns and their derived displacements. We can interpret $\boldsymbol{F} = \mathcal{F}\boldsymbol{\lambda}$ to be the change in the representation of the loads, where $\boldsymbol{\lambda}$ is the intensity of a particular load combination in a space that is restricted to the capabilities of the $\mathcal{F}$ representation.

At this point, we can search for the most unfavourable format for those patterns. In other words, we search for the load patterns $\mathcal{F}$ that generate the maximum loss of potential energy for a given load intensity. Henceforth, we consider $\mathcal{F}$ to be solely a column with one such pattern and with a known measure: $\|\mathcal{F}\| = \mathfrak{f}^2$ (we will use the sum of squares norm: $\|\mathcal{F}\| = \mathcal{F}^T\mathcal{F}$). Hence, the problem would consist in solving for $\max_\mathcal{F} \mathcal{U}^T\mathcal{F}$ or $\max_\mathcal{F} \mathcal{F}^T K^{-1}\mathcal{F}$ for the normalized equivalent $\mathcal{F}$ patterns. In this way, we are searching for the most flexible responses of the structure for a given load intensity.

The problem can be stated as the maximization of the functional $\mathfrak{F}$, via the Lagrange multipliers.

$$\max_\mathcal{F} \mathfrak{F} = \max_\mathcal{F} \left( \mathcal{F}^T K^{-1}\mathcal{F} - \frac{1}{\lambda}(\mathcal{F}^T\mathcal{F} - \mathfrak{f}^2) \right). \qquad (4.12)$$

This problem can be easily solved with

$$\frac{\partial \mathfrak{F}}{\partial \mathcal{F}} = 2\left(K^{-1}\mathcal{F} - \frac{1}{\lambda}\mathcal{F}\right) = \mathbf{0}, \tag{4.13}$$

$$\left(K^{-1} - \frac{1}{\lambda}I\right)\mathcal{F} = \mathbf{0} \tag{4.14}$$

and
$$(K - \lambda I)\mathcal{U} = \mathbf{0}. \tag{4.15}$$

Here, we have two eigenvalue problems that can be easily related in cases where $K$ is invertible.

This last problem could be simplified to find $\mathcal{F}$ patterns parallel to their respective $\mathcal{U}$ displacements, as in $\mathcal{F} = \lambda \mathcal{U}$ which leads us to the problem of solving situations where

$$\left.\begin{array}{r}(K - \lambda I)\mathcal{U} = \mathbf{0} \\ |K - \lambda I| = 0\end{array}\right\} \tag{4.16}$$

and

that are similar to those that have been previously considered.

The problem can be compared to the diagonalization of the flexibility or the stiffness matrix, where each of the load–displacement vectors responds to one from among a collection of several uncorrelated deformation modes (in fact, an orthogonal set of force-deformation modes). The eigenvalue $1/\lambda$ is a value that ensures the following for each eigenvector $\mathcal{F}$: that the surfaces representing the energy loss $\mathcal{F}^T K^{-1} \mathcal{F}$ and the *distance* state of the forces $\mathcal{F}$ that meet the requirement $\mathcal{F}^T \mathcal{F} - \hat{f}^2$, are represented in a common scale where $\lambda(\mathcal{F}^T \mathcal{F} - \hat{f}^2)$, and are tangent. $1/\lambda$ is the flexibility associated with the corresponding $\mathcal{F}$ eigenvector and $\lambda$ is the stiffness associated with the corresponding $\mathcal{U}$ eigenvector.

The diagonalization of the stiffness matrix allows us to perform the decomposition of each load–displacement problem as the superposition of such elemental modes, from the most to the least significant. As a classical example, we can consider the superposition of the symmetric and antisymmetric components of a load in a symmetric structure.

It is well known that for this last eigenvalue problem equations (4.15) and (4.16) permit us to find the internal mechanisms for a singular condition in the stiffness matrix that corresponds to the eigenvectors related to the null $\lambda$ eigenvalues; i.e. in situations where non-null $U$ displacements would be associated with null $F$ external forces. This provides an indirect way to obtain a basis for the right null space $N$ of the compatibility matrix $B$.

## 4.5. Dimensionality or noise reduction in structural models

The application of SVD, or eigenvalues-eigenvectors decomposition in the case of a symmetric matrix, aiming to reduce the dimensionality of the original model is a key technique that has been widely used in dynamic analysis to reduce the vibration modes to those that have more influence, the higher modes.

This reduction is largely related to the relative weight of the singular values; these contain the weight of the information corresponding to their respective form vectors (eigenvectors or left and right vectorial spaces in the case of SVD). In information science, this reduction is usually of the order of several orders of magnitude, as is the case in the reduction of dynamics.

The reduction substitutes the matrix that represents the linear map with its reduced approximation. This procedure eliminates the least significant and near null components from the representation, in addition to both null spaces of the problem.

$$A = v\Sigma V^T \tag{4.17}$$

and

$$A \approx v_r \Sigma_r V_r^T, \tag{4.18}$$

where $r$ is the number of elements selected for the reduced basis. And where the measure of the approximation can be controlled by the number of elements in the diagonal of $\Sigma$ needed to obtain a significant part of its total trace (sum of diagonal elements), using to evaluate this measure the quotient $\mathrm{trace}(\Sigma_r)/\mathrm{trace}(\Sigma)$.

At the present day, there are several excellent methods by which technical problems can be solved. For example, solving a structural model with a large number of unknowns through its modelling and analysis with the support of a standard finite-element software. However, this creates the risk of over-

representing the level of detail for a problem, which increases the resources allocated to solving problems that could be addressed by simpler means. Consequently, the capability of perceiving the significance of the different variables involved in problems is being progressively compromised, reducing the opportunities to develop improved or innovative solutions. In this scenario, resorting to the dimensional reduction provided by the studied techniques could be more useful than it first appears. For example, the flexibility reduction described in the next section can be approached from the perspective of load-eigenvalues, and the majority of the problematic behaviours can be easily condensed into two of the most flexible modes.

This approach is not only an interesting pedagogic tool but is also an excellent instrument for the application of extended theoretical developments, and could be used in the analysis of structural behaviour in stochastic scenarios.

# 5. Applicability of the dualities and their related techniques

The following example illustrates the concepts described in the previous sections.

## 5.1. Frame

In this example (see figure 1), we chose a simple frame with vertical and horizontal loads. We add a tension diagonal bar and a slanted support to highlight some concepts that would not be evident in orthogonal geometries. In addition, the joint between the support and the beam is assumed to add forced (prestress) movements to the assemblage, which add to the possible prestressing effects on the diagonal.

For ease of comparison, we use loads that are the product of a base load and a load factor. This results in an equal base load (unfactored load) for the comparison of the equilibrium component of the solutions, and different load factors for the comparisons between approaches (limit load, deflections, member requirements and so on).

## 5.2. Geometry and basic variables

The frame has a span length of $l = 10$ m, a height $h = 4$ m, a slant given by $s = 0.3$ m and a cantilever with length $c = 1.5$ m. Loads are given by $F_h$ (wind), in joint 1, and 0.8 $F_v$, $F_v$, 0.8 $F_v$ and 0.5 $F_v$ (gravity) in joints 1, 2, 3, 4, respectively, where joint 2 is in the centre of the span. The seismic horizontal forces are applied to the same points where the mass corresponding to the gravitational forces are considered to be. The frame will be constructed using S325 HEB supports and an IPE beam, with a diagonal made of S500 steel.

Joints are considered to be reinforced and thus rigid and sufficiently resistant. The deformation and failure conditions are constrained to the ends of the members. For most of the calculations we assume $20 \times 40$ cm$^2$ as the approximate boundary dimensions for the joints. Axial elongations in the columns and the beam are disregarded.

Figures 2–6 represent the positive *independent* movements. In addition, figures 2 and 6 describe the movements at joint 3 ('horizontal' movement and rotation) in detail. Since the cantilever is an isostatic element, only its static resultants are included in the model.

Thus, $U$ is represented by a column vector which corresponds to a load vector $F$
$$U^T = [U_1, U_2, U_3, U_4, U_5] = [\Delta_h, \Delta_{v2}, \theta_1, \theta_2, \theta_3]$$
and
$$F^T = [F_1, F_2, F_3, F_4, F_5]$$
that is the work conjugate of $U$ (i.e. $W = U^T F$). This condition imposes constraints on the components of $F$, which are given by the following values depending on the load case
$$F_g^T = [(F_{v3} + F_{v4}) \tan \alpha, F_{v2}, 0, 0, -F_{v4}c]$$
$$= [(0.8 + 0.5)F_V \tan \alpha, F_v, 0, 0, -(0.5F_v)c]$$
for the gravitational case,
$$F_w^T = [F_h, 0, 0, 0, 0]$$
for the wind focused case, and

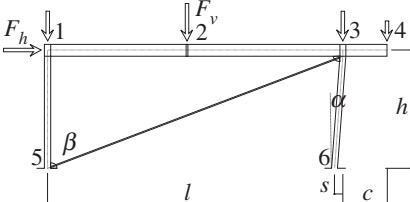

**Figure 1.** Frame geometry.

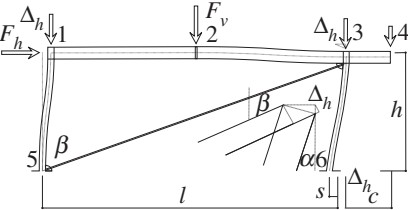

**Figure 2.** Displacement 1: 'horizontal' displacement.

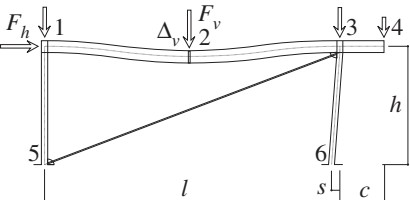

**Figure 3.** Displacement 2: vertical displacement of joint 2.

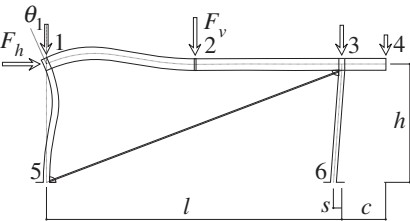

**Figure 4.** Displacement 3: rotation of joint 1.

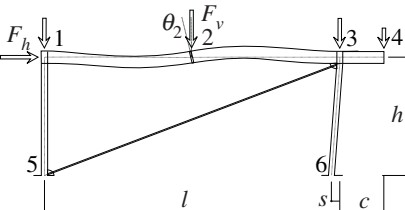

**Figure 5.** Displacement 4: rotation of joint 2.

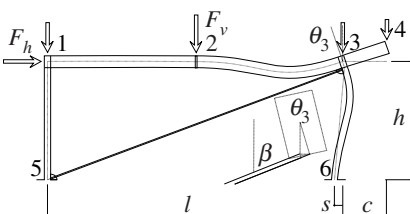

**Figure 6.** Displacement 5: rotation of joint 3.

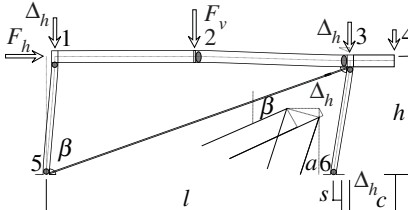

**Figure 7.** Plastic horizontal displacement.

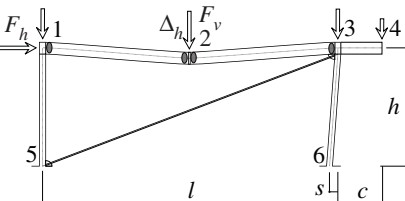

**Figure 8.** Plastic vertical displacement of joint 2.

$$\mathbf{F}_s^T = [(F_{v3} + F_{v4})\tan\alpha + \alpha_s(F_{v1} + F_{v2} + F_{v3} + F_{v4}),$$
$$F_{v2}, 0, 0, -F_{v4}c]$$
$$= [(1.3)F_v\tan\alpha + \alpha_s(3.1)F_v, F_v, 0, 0, -(0.5F_v)c]$$
$$\text{for the seismic and gravitational case,}$$

where $\alpha_s$ is the seismic base shear factor.

Each load or external force component in the vector $\mathbf{F}$ can be calculated as the work done by the applied loads in the corresponding unitary displacement for each case. Therefore, there are 'horizontal' components $F_1$ derived from vertical loads. These components are the result of the constant axial length of the slanted support, which imposes a vertical displacement on joint 3 with the value $\Delta_h\tan\alpha$, where $\Delta_h$ is the horizontal displacement.

Here, we consider the following to be the deformations $\mathbf{u}$: the end rotations in supports and beams relative to the straight line between both displaced ends; and the total elongation in the case of the diagonal. The end rotations are considered to be the same positive signs as those selected for the joints.

$$\mathbf{u}^T = [\theta_{11}, \theta_{12}, \theta_{21}, \theta_{22}, \theta_{51}, \theta_{52}, \theta_{61}, \theta_{62}, \delta].$$

The bars are denoted by the joint at its first end. Hence, bar 1 is the part of the beam between joints 1 and 2, while bar 2 is between joints 2 and 3; bars 5 and 6 are the supports. The internal forces are those exerted by the joints onto the end of the members, i.e. the forces are the end moments in the bars (supports as well as the two different parts of the beam). These internal forces have positive signs that are consistent with the positive signs of the rotations. The axial (normal) force in the diagonal will have a positive sign if it is under tension.

$$\mathbf{f}^T = [m_{11}, m_{12}, m_{21}, m_{22}, m_{51}, m_{52}, m_{61}, m_{62}, n_d].$$

Here, we will consider the representation of the limit analysis. In this case, the parameters will be the same with one exception: for the beams and columns the end rotations will be derived from concentrated hinges located at their ends, while the elongation for the diagonal is derived from a concentrated striction located somewhere in the bar. Hence, we can employ the same global $\mathbf{U}$, $\mathbf{F}$ and local $\mathbf{u}$, $\mathbf{f}$ parameters. Figures 7 and 8 show the plastic interpretation for the horizontal and the vertical displacements.

## 5.3. Compatibility and equilibrium equations

The compatibility equation $\mathbf{u} = \mathbf{BU}$ can be easily derived from the figures that describe the independent global movements (2) to (6). Furthermore, if we consider that the ends of the bars are

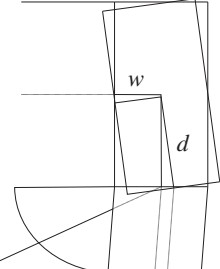

**Figure 9.** The rotation of joint 3 in detail.

located in the joints coordinates, with the exception of the eccentricity of the diagonal $d$, whose depth is half the beam, we have

$$
\begin{bmatrix}
\theta_{11} \\
\theta_{12} \\
\theta_{21} \\
\theta_{22} \\
\theta_{51} \\
\theta_{52} \\
\theta_{61} \\
\theta_{62} \\
\delta
\end{bmatrix}
=
\begin{bmatrix}
0 & \frac{1}{l_1} & 1 & 0 & 0 \\
0 & \frac{1}{l_1} & 0 & 1 & 0 \\
\frac{\tan\alpha}{l_2} & -\frac{1}{l_2} & 0 & 1 & 0 \\
\frac{\tan\alpha}{l_2} & -\frac{1}{l_2} & 0 & 0 & 1 \\
\frac{1}{l_5} & 0 & 0 & 0 & 0 \\
\frac{1}{l_5} & 0 & 1 & 0 & 0 \\
\frac{1}{\cos\alpha\, l_6} & 0 & 0 & 0 & 0 \\
\frac{1}{\cos\alpha\, l_6} & 0 & 0 & 0 & 1 \\
\sin\beta - \tan\alpha\cos\beta & 0 & 0 & 0 & d\sin\beta
\end{bmatrix}
\begin{bmatrix}
\Delta_h \\
\Delta_{v2} \\
\theta_1 \\
\theta_2 \\
\theta_3
\end{bmatrix}.
$$

Here, $l_i$ is the length of the bar $i$ between the ends supposed located in the joint coordinates, excluding the diagonal. However, if we assume huge rigidity and resistance of the joints, the movements and forces of the members must be referred to their ends, located in the surface of the contact to the joints. Hence, those contacts are the reference for the measurements of the lengths. The compatibility equations must be reformulated to account for eccentricities $[w, d]$, which correspond to half the width or half the depth of the joint in both joints 1 and 3.

$$
\begin{bmatrix}
\theta_{11} \\
\theta_{12} \\
\theta_{21} \\
\theta_{22} \\
\theta_{51} \\
\theta_{52} \\
\theta_{61} \\
\theta_{62} \\
\delta
\end{bmatrix}
=
\begin{bmatrix}
0 & \frac{1}{l_1} & 1+\frac{w}{l_1} & 0 & 0 \\
0 & \frac{1}{l_1} & \frac{w}{l_1} & 1 & 0 \\
\frac{\tan\alpha}{l_2} & -\frac{1}{l_2} & 0 & 1 & \frac{w}{l_2} \\
\frac{\tan\alpha}{l_2} & -\frac{1}{l_2} & 0 & 0 & 1+\frac{w}{l_2} \\
\frac{1}{l_5} & 0 & \frac{d}{l_5} & 0 & 0 \\
\frac{1}{l_5} & 0 & 1+\frac{d}{l_5} & 0 & 0 \\
\frac{1}{\cos\alpha\, l_6} & 0 & 0 & 0 & \frac{d}{\cos\alpha\, l_6} \\
\frac{1}{\cos\alpha\, l_6} & 0 & 0 & 0 & 1+\frac{d}{\cos\alpha\, l_6} \\
\sin\beta - \tan\alpha\cos\beta & 0 & 0 & 0 & d\sin\beta
\end{bmatrix}
\begin{bmatrix}
\Delta_h \\
\Delta_{v2} \\
\theta_1 \\
\theta_2 \\
\theta_3
\end{bmatrix}.
$$

Here, we have disregarded the slight lowering of joint 3 during its rotation, that results from the null change in length of the slanted support (bar 6), see figure 9. From both compatibility equations considered, we will continue with the last one.

Equilibrium equations can be constructed by the transposition of **B**. Alternatively, we can consider the equilibrium equation for the work associated with each independent movement. Figure 10 shows the construction of said equation in the most complex case: the case for the 'horizontal' forces. This case is complex because the slanted bar 6 cannot be elongated, and thus requires the existence of axial forces to maintain vertical equilibrium. As these forces are oblique ($v_a$, $v_b$, $v_c$ in the same figure), horizontal components must be added to maintain this equilibrium. From this figure, we can also obtain the transpose of the first column of **B** as the first row of **H**, since the shear forces are $T_i = (m_{i1} + m_{i2})/l_i$ and the axial components have an angle of $\alpha$ with the vertical.

## 5.4. Constitutive or material equations

In elastic conditions, these are the relationships between the end forces and the end displacements, i.e. the stiffness matrices of the bars. Here, we have the classic form of the flexural stiffness for the beams and

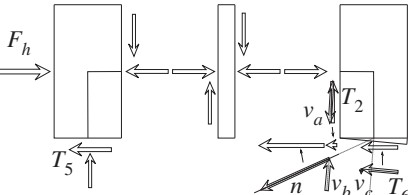

**Figure 10.** Horizontal equilibrium of forces.

supports, and the axial stiffness for the diagonal

$$k_i = \frac{E_i I_i}{l_i} \begin{bmatrix} 4 & 2 \\ 2 & 4 \end{bmatrix} \quad k_d = \frac{E_d A_d}{l_d},$$

where the matrix $k$ is constructed as a blocked diagonal matrix, as the stiffness is uncoupled between its members

$$k = \begin{bmatrix} k_1 & 0 & 0 & 0 & 0 \\ 0 & k_2 & 0 & 0 & 0 \\ 0 & 0 & k_5 & 0 & 0 \\ 0 & 0 & 0 & k_6 & 0 \\ 0 & 0 & 0 & 0 & k_d \end{bmatrix}.$$

At plastic or limit conditions, the constitutive equations are now the resistance conditions. Hence, we can assign the following common resistances according to the distinct elements: the flexural resistance $r_b$ to all the beam ends (for both positive and negative moments); the resistance $r_s$ to all the support ends, and the axial resistance $r_d$ to the diagonal, which works only under tension. These resistance conditions can be written as in equation (2.11)

$$\begin{bmatrix} 1 & 0 & 0 & 0 & 0 & 0 & 0 & 0 & 0 \\ -1 & 0 & 0 & 0 & 0 & 0 & 0 & 0 & 0 \\ 0 & 1 & 0 & 0 & 0 & 0 & 0 & 0 & 0 \\ 0 & -1 & 0 & 0 & 0 & 0 & 0 & 0 & 0 \\ 0 & 0 & 1 & 0 & 0 & 0 & 0 & 0 & 0 \\ 0 & 0 & -1 & 0 & 0 & 0 & 0 & 0 & 0 \\ 0 & 0 & 0 & 1 & 0 & 0 & 0 & 0 & 0 \\ 0 & 0 & 0 & -1 & 0 & 0 & 0 & 0 & 0 \\ 0 & 0 & 0 & 0 & 1 & 0 & 0 & 0 & 0 \\ 0 & 0 & 0 & 0 & -1 & 0 & 0 & 0 & 0 \\ 0 & 0 & 0 & 0 & 0 & 1 & 0 & 0 & 0 \\ 0 & 0 & 0 & 0 & 0 & -1 & 0 & 0 & 0 \\ 0 & 0 & 0 & 0 & 0 & 0 & 1 & 0 & 0 \\ 0 & 0 & 0 & 0 & 0 & 0 & -1 & 0 & 0 \\ 0 & 0 & 0 & 0 & 0 & 0 & 0 & 1 & 0 \\ 0 & 0 & 0 & 0 & 0 & 0 & 0 & -1 & 0 \\ 0 & 0 & 0 & 0 & 0 & 0 & 0 & 0 & 1 \\ 0 & 0 & 0 & 0 & 0 & 0 & 0 & 0 & -1 \end{bmatrix} \begin{bmatrix} m_{11} \\ m_{12} \\ m_{21} \\ m_{22} \\ m_{51} \\ m_{52} \\ m_{61} \\ m_{62} \\ n_d \end{bmatrix} \leq \begin{bmatrix} 1 & 0 & 0 \\ 1 & 0 & 0 \\ 1 & 0 & 0 \\ 1 & 0 & 0 \\ 1 & 0 & 0 \\ 1 & 0 & 0 \\ 1 & 0 & 0 \\ 1 & 0 & 0 \\ 0 & 1 & 0 \\ 0 & 1 & 0 \\ 0 & 1 & 0 \\ 0 & 1 & 0 \\ 0 & 1 & 0 \\ 0 & 1 & 0 \\ 0 & 1 & 0 \\ 0 & 1 & 0 \\ 0 & 0 & 1 \\ 0 & 0 & 0 \end{bmatrix} \begin{bmatrix} r_b \\ r_s \\ r_d \end{bmatrix},$$

where the components $\psi$, $f$, $d$, $r$ can be immediately identified.

## 5.5. Elastic solution

We can address two different problems through the elastic approach: analysis problems and design problems.

For the first class of problem, the analysis problems, the dimensions for all the bars must be chosen. Thus, we have assumed the following: HEB200 for the supports ($I_y = 5696$ cm$^4$, $W_{py} = 642.5$ cm$^3$), and IPE400 for the beam ($I_y = 23130$ cm$^4$, $W_{py} = 1307$ cm$^3$), both of which are made of steel and have a yielding limit of 325 MPa; and a steel tendon $\phi 25$ ($A = 4.909$ cm$^2$) with a yielding limit of 500 MPa for the diagonal. Each has an elastic modulus of $E = 200$ GPa. This results in a stiffness matrix $K = B^T k B$ where the length dimensions are measured in dm and the force dimensions are measured in kN. It is important to note that the use of dm is deliberate: in doing so, the numerical values for both, areas

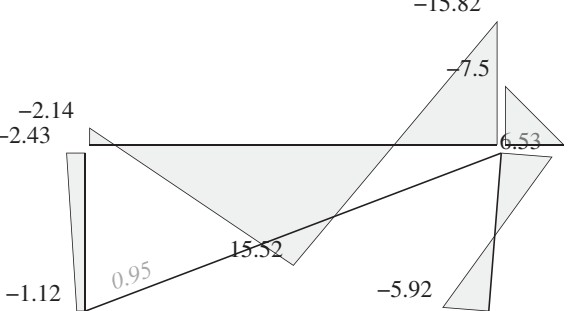

**Figure 11.** Bending moments in the bars (outside joints) measured in dm kN, for unit wind and gravitational loads in the elastic eccentric model. The label 0.95 represents the axial tension in kN.

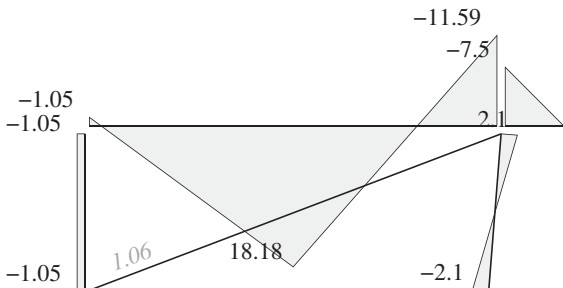

**Figure 12.** Approximate bending moments in devised equilibrium solution.

and inertias, are of the same order of magnitude. Given that the areas and inertias are combined into a single matrix, this choice reduces the numerical round-off noise.

$$
\boldsymbol{K} = \frac{E}{100}
\begin{bmatrix}
0.06317 & -0.00186 & 0.26159 & 0.04563 & 0.38699 \\
-0.00186 & 0.04718 & 0.60160 & 0.00000 & -0.60160 \\
0.26159 & 0.60160 & 27.05356 & 10.01883 & 0.00000 \\
0.04563 & -0.00000 & 10.01883 & 37.76327 & 10.01883 \\
0.38699 & -0.60160 & 0.00000 & 10.01883 & 27.19427
\end{bmatrix}.
$$

Now, we can obtain the elastic results (figure 11). We used unitary values in both $F_v$ and $F_h$ to describe the superposition of gravitational and wind cases in terms of bending moments. It can be seen that the worst condition relative to resistance is reached at the index 9 in the list of internal forces, i.e. in $n_d$. For a simultaneous increment of $F_v$ and $F_h$, this represents a load factor of value $\gamma_e = f_y A_d / n_d = 245.45/0.95 = 258$. A classic analysis must also take into account lateral and vertical deflections, as they must be compared with their respective admissible limits. This procedure is straightforward. However, in the case where the predefined design does not satisfy all the requirements, it must be modified and a new analysis step must be performed. This constitutes the most usual case where lateral drift is a limiting condition.

In design problems, we must choose an equilibrium solution and a compatible displacement-deformation field, and select sections that simultaneously satisfy both conditions in the MAT equations.

— Firstly, we choose an *arbitrary* (approximate) equilibrium solution, as shown in figure 12. Here, we decided to assign 0.9 of the horizontal resultant to the diagonal, an arbitrary but sensible decision. Thus we have a tension of 1.06 kN in the diagonal and plausible bending moments in the bars required to equilibrate the remaining horizontal and vertical loads.

— Later, we can design an *arbitrary* (approximate) compatible deformation solution as shown in figure 13. In this scenario, the horizontal displacement is predefined as 1/300 of the height of the frame and the vertical deflection is limited to 1/400 of the span of the frame. Furthermore, the rotations at joints 1, 2 and 3, are estimated to be consistent with the dimensions of the bars and the horizontal displacement induced by the diagonal elongation, assuming that the bars are of constant section and obey bending diagrams as shown.

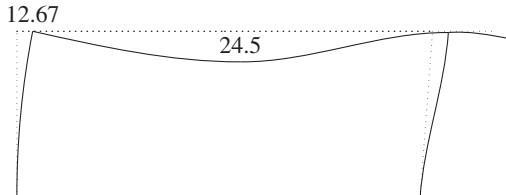

**Figure 13.** Approximate compatible deformation selected for the design. Movements have been amplified 30 times. Displacements in mm.

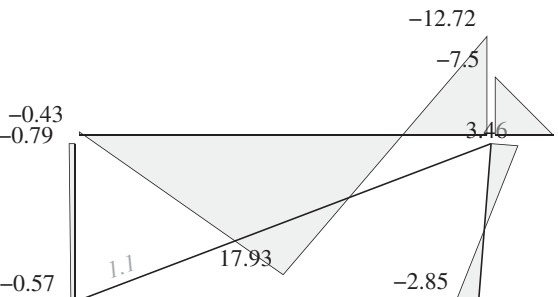

**Figure 14.** Exact bending moments in the bars (outside joints) measured in dm kN, for the wind and gravity loads, calculated for the frame devised in the elastic design problem.

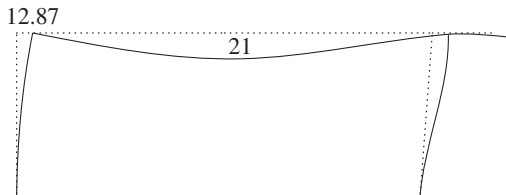

**Figure 15.** Exact deformation calculated for the frame devised in the elastic design problem, movements amplified 30 times. Displacements in mm.

As mentioned before, both solutions must exhibit deformations and internal forces with signs that are consistent with each other.

Therefore, the selection of the sections necessary to meet all the requirements that exist for each element in terms of internal forces and deformations is an immediate consequence of the material constitutive conditions. In order to attain with the design a load factor equal to that found before for the elastic solution ($\gamma_e = 258$), the following properties are needed for the elements: a section of 12.22 cm$^2$ for the diagonal, and inertias of 5167 cm$^4$ for the supports and 76 616 cm$^4$ for the beam. These results would require a design with HE200B supports and a HE450B (or an excessive IPE600) beam.

The elastic analysis of this design generates the results illustrated in figure 14 for unscaled loads. It exhibits an axial tension of 1.1 kN in the diagonal, and has a horizontal displacement that is virtually equal to the value previously selected for the design, with a vertical deflection of 1/473 of the span of the beam; both movements corresponding to the case of the factored loads, as shown in figure 15.

The differences between the devised and the exact equilibrium solutions come from the rough approximations selected for equilibrium and deformation, and thus the slight inconsistencies between them for a design with constant sections.

## 5.6. Limit solution

We can address the same two problems that we tackled with the elastic approach by using a limit solution.

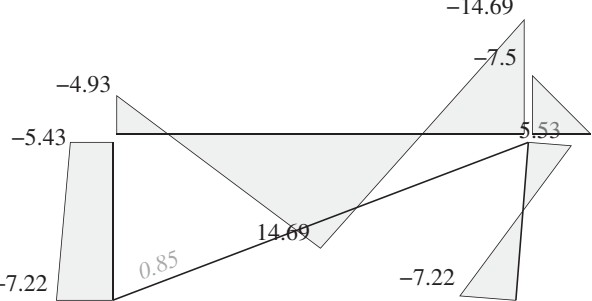

**Figure 16.** Static plastic analysis results for the given loads and members.

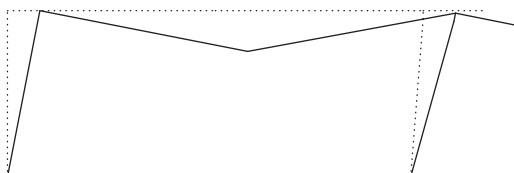

**Figure 17.** Collapse geometry, derived from the plastic kinematic analysis.

For the analysis problems, we establish the conditions as described in equations (2.14) or (2.15). After employing the corresponding yielding limit to each class of steel, we obtain the load factor $\gamma_p = 289.18$ for the same problem that was solved using the elastic approach (with lpSolve [21]). It results in the bending moments given in figure 16 that equilibrate the unscaled loads and generate a tension of 0.85 kN in the diagonal. These results signify that the plastic limits are attained at the following points: the diagonal, the hinges at the centre of the beam, and in three of the ends of the supports. This result can be assessed by both the static and the kinematic approaches.

The kinematic approach not only gives us the collapse geometry $\dot{U}$ (figure 17) but also the $\lambda$ multipliers that describe the resistance limits that have been attained. Other elements of importance such as the components of the plastic flow $\dot{u}$, can be obtained with equation (2.13).

For design problems, we first devise an equilibrium solution and adopt sections strong enough to resist the devised internal forces. In this way, we obtain a solution that is supported by the static theorem. However, the optimum limit design problem is more interesting. Here, we must state the linear cost functions for the base strengths $r^T = [r_b, r_s, r_d]$ (beam, supports and the diagonal). To do so, we have to approximate the coefficients of the linear regression of the areas of the selected families of sections from the values of these resistance variables. Thus, taking into account the plastic modulus of IPEs ranging from 300 to 450 or of HEBs ranging from 160 to 300, and the area of the diagonal, and disregarding the constant terms in those regressions, we obtain: $l^T = [0.418, 0.62, 1]$. Furthermore, if we multiply each of those coefficients by the total lengths of each section $t^T = [115, 76.118, 105.109]$ (dm) and by the unitary costs per volume of the section type $c_{uV}^T = [0.9, 1, 0.8]$ we obtain the cost vector $c$, where each index is $c_i = l_i t_i c_{uv,i} : c^T = [43.223, 47.196, 84.088]$. Equation (2.26) can thus be stated and solved for loads scaled by the same load factor as the one calculated for the elastic solution ($\gamma_e$). As before, lpSolve requires positive decision variables that lead to the following form:

$$\min \left( \begin{bmatrix} \mathbf{0}^T & \mathbf{0}^T & c^T \end{bmatrix} \begin{bmatrix} f_p \\ f_n \\ r \end{bmatrix} \right)$$

$$\begin{bmatrix} \boldsymbol{\psi} & -\boldsymbol{\psi} & -d \\ B^T & -B^T & \mathbf{0} \end{bmatrix} \begin{bmatrix} f_p \\ f_n \\ r \end{bmatrix} \begin{matrix} \leq \\ = \end{matrix} \begin{bmatrix} \mathbf{0} \\ F \end{bmatrix}$$

$$f_p \geq \mathbf{0}; \, f_n \geq \mathbf{0}, \, r \geq \mathbf{0}$$

$$f = f_p - f_n.$$

In the optimal solution, the area required for the diagonal is 4.94 cm², and the plastic modulus for the beam and the columns are 1164 and 429 cm³, respectively; these values are very close to IPE400 for the

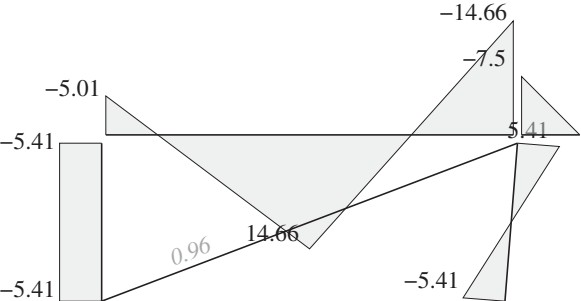

**Figure 18.** Bending moments in the bars for the unfactored loads in the optimal plastic solution with a load factor of 258.

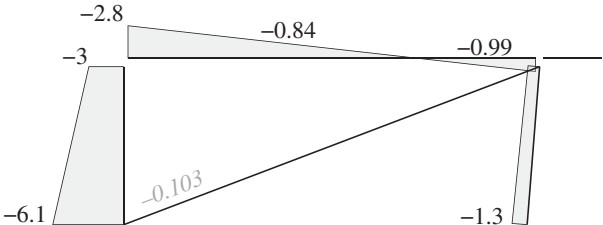

**Figure 19.** Bending moments for the (unfactored) self-stress state reached via ductile flow.

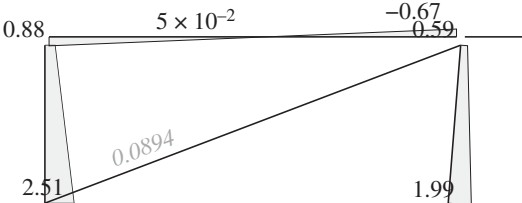

**Figure 20.** Bending moments for self-stress base state no. 1.

beam and HEB180 for the supports. The resulting equilibrium solution for unfactored loads is shown in figure 18.

## 5.7. Ductility requirements

As stated in equations (3.15) and (3.17), we can obtain the ductility requirements of each member by computing the difference between the plastic and the elastic analysis problems. Hence, we compute the difference for the original frame calculated in §5.5 and 5.6. In the case of the unitary values of $F_v$ and $F_h$, the forces in the members are illustrated in figures 11 ($f_{u,e}$) and 16 ($f_{u,p}$). Their factored differences can be interpreted via equations (3.9) and (3.10), with $f_e = \gamma_e f_{u,e}$ and $f_p = \gamma_p f_{u,p}$. Thus $\gamma$ in equation (3.10) is $\gamma = \gamma_p/\gamma_e$ and $f_d = \gamma_p(f_{u,p} - f_{u,e})$, where $f_d = Du_d$ is a set of internal forces that is shown unfactored ($f_{u,p} - f_{u,e}$) in figure 19. We can verify that this set belongs to the self-stress subspace, i.e. the set of internal forces in self-equilibrium: $B^T f_d = 0$ (see discussion on equation (3.13)).

In order to find the $u_d$ requirements, equation (3.15), we have to perform the following steps:

— Firstly, we solve the $D$ eigenvalue problem equation (3.12) obtaining the incompatible deformation base $u_r$ for the self-stress subspace of $u$, which has a dimension of $4 = 9 - 5$ (dimension of $u$ space − dimension of $U$ space), and the null base $u_n$. This incompatible base can be viewed from the perspective of incompatible deformation, as well as from the independent possible prestress states that follow from those deformations. These stress states are shown (in an arbitrary scale) in figures 20–23.

— Secondly, we find the coefficients $g$ for the influence of the incompatible base $u_r$, as derived from equation (3.17).

— Thirdly, we find the coefficients $\alpha$ for the influence of the null base $u_n$, by selecting from the plastic kinematics solution the $\dot{u}_i$ deformations that include the four deformations with null flow plus the deformation with the least energy, that would correspond to the last hinge, and by applying equation (3.19).

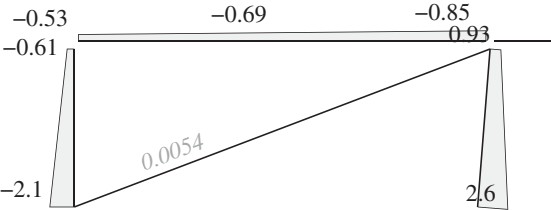

**Figure 21.** Bending moments for self-stress base state no. 2.

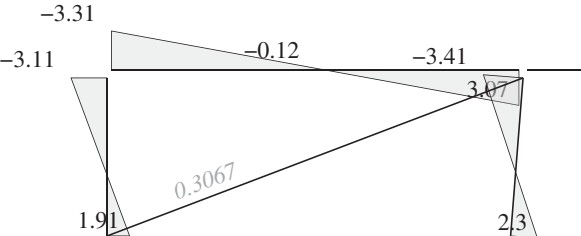

**Figure 22.** Bending moments for self-stress base state no. 3.

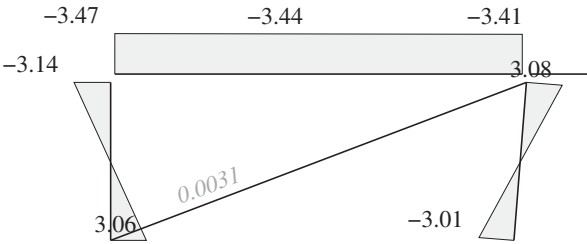

**Figure 23.** Bending moments for self-stress base state no. 4.

Furthermore, the ductilities required, also known as the demand ductilities, must be compared with the extensional or the rotational ductility capacity of the members, which are defined in tables 5–6 and 5–7 in ASCE–41 or in the tables B1–B4 of the Eurocode 8 part 3. The results are shown in table 2.

In cases where it is not possible to determine the last hinge, it is necessary to explore all the possible positions to derive for each position the results at the beginning of the collapse. Table 3 illustrates this methodology for the example presented here.

We can see that in the solutions where the location of the last hinge is incorrectly selected, there exist contradictions between the $u_i$ deformations required and tabulated in the corresponding row and the corresponding internal forces ($f_{pi}$) signs. Moreover, the global plastic energy $W_i$ is maximum for the solution where the last hinge ($i$) is correctly selected (as obtained from equation (3.20), and shown in mkN in the table).

Having calculated the internal forces $f$, of which some are limit forces, we can calculate the associated elastic deformations $u_e$. These are deformations that, when added to the ductile deformations got in the plastic flow, $u_d$, provide a compatible set $u = u_e + u_d$, from which we can derive the displacements $U$ attained at the beginning of the collapse by applying equation (4.11).

Figure 24 illustrates this movement and deformation.

## 5.8. Global limit conditions

Here, we present the restricted subspaces of $R$ and $\Lambda$ spaces derived from two independent collapse modes $\Lambda_1$, $\Lambda_2$ and their combined $\Lambda_3$ mode in order to illustrate equations (2.17) to (2.23)

$\Lambda_1$ Failure of the columns and the diagonal. This mode is related to horizontal displacement.

$\Lambda_2$ Failure at the centre of the beam with no horizontal displacement in the left column, with compatible hinges at the upper ends of both columns, and with a lower hinge at the right column and a diagonal extension, both of which are required by the eccentricities.

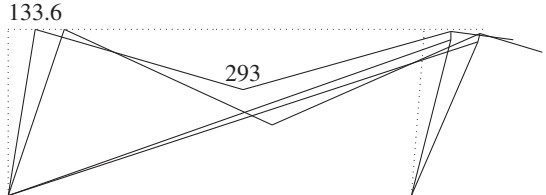

**Figure 24.** The start and the early stages of collapse geometries, where the second stage selects an elongation in the diagonal that doubles that of the start. Displacements in mm for the start. Figure includes elastic and plastic deformations amplified five times.

**Table 2.** Solution for the ductility requirements problem: coefficients for the incompatible and null base components and the resulting ductile flow.

| order | 1 | 2 | 3 | 4 | 5 |
|---|---|---|---|---|---|
| **g** | $g_1$ | $g_2$ | $g_3$ | $g_4$ | |
| | −0.028 | 0.012 | 0.003 | 0.001 | |
| **α** | $\alpha_1$ | $\alpha_2$ | $\alpha_3$ | $\alpha_4$ | $\alpha_5$ |
| | 0.078 | −0.028 | 0.87 | −0.221 | −0.01 |
| $u_d$ | $u_2$ rad | $u_4$ rad | $u_7$ rad | $u_9$ dm | |
| | 0.097 | −0.078 | 0.019 | 0.893 | |

**Table 3.** Ductility requirements for arbitrary locations of last hinge (the one with null flow in each row) selected from among the non-null deformations in the plastic solution. Included are the internal plastic forces in each member and global plastic energy $W_i$ (in mkN) for each assumption, as described by equation (3.20).

| $u_2$ rad | $u_4$ rad | $u_5$ rad | $u_7$ rad | $u_9$ dm |
|---|---|---|---|---|
| 0 | 0.019 | −0.049 | −0.03 | −0.785 |
| 0.019 | 0 | −0.04 | −0.02 | −0.463 |
| 0.097 | −0.078 | 0 | 0.019 | 0.893 |
| 0.059 | −0.04 | −0.019 | 0 | 0.23 |
| 0.045 | −0.027 | −0.026 | −0.007 | 0 |
| $f_{p2}$ mkN | $f_{p4}$ mkN | $f_{p5}$ mkN | $f_{p7}$ mkN | $f_{p9}$ kN |
| 424.78 | −424.78 | 208.81 | 208.81 | 245.45 |
| $W_2$ | $W_4$ | $W_5$ | $W_7$ | $W_9$ |
| −43.574 | −15.975 | 100.292 | 43.455 | 23.736 |

$\Lambda_3$ Superposition of both modes of failure in which we obtain a failure condition that includes a hinge at the upper end of the left column.

Those collapse modes can be tabulated so as to represent the correspondence between the indices for the collapse mode and the local plastic deformation, as shown in the following table.

|  | $u_1$ | $u_2$ | $u_3$ | $u_4$ | $u_5$ | $u_6$ | $u_7$ | $u_8$ | $u_9$ |
|---|---|---|---|---|---|---|---|---|---|
| $\Lambda_1$ | 0 | 0 | 0 | 0 | 1 | 1 | 1 | 1 | 1 |
| $\Lambda_2$ | 0 | 1 | 0 | 0 | 0 | 1 | 1 | 1 | 1 |
| $\Lambda_3$ | 0 | 1 | 0 | 0 | 1 | 0 | 1 | 1 | 1 |

where it can be seen that the number of zeros in each row is equal to $N - 1$. This is because there are $n - N$ hyperstatic conditions, and consequently, $n - N + 1$ plastic conditions that need to be attained so as to produce the collapse, with possible local collapse modes. This implies that the remaining $n - (n - N + 1) = N - 1$ deformations are null.

To construct the geometry of the aforementioned modes (for positive deformations in our restricted subspace), we must perform the following steps:

— Firstly, we must obtain the compatible deformations $\dot{u}_j$ that correspond to the indices in the last table for each $\Lambda_i$ collapse mode. These deformations are obtained from the compatible deformations base $v$ calculated in the SVD of the matrix $B$: $B = v\Sigma V^T$. This approach is similar to the methods employed to solve the ductility problem.

  — Obtain from the SVD a reduced base with five components.
  — Pose the four equations to establish the four known null deformations.
  — Pose an additional equation required to obtain the five coefficients for the combination of that base. This equation is the normalization: $\dot{u}^T f = \dot{U}^T F = \lambda^T r = \Lambda^T R = 1$, where $f$ is derived from the strength equations (2.11) that are relevant in such a collapse mode; the relevant $\psi$ columns are those that are associated with the non-null $u$ deformations, and the relevant rows are those with positive values in those columns, as we are restricting our search to collapse modes that have positive components in $u$, or $f$. Thus, the collapse forces $f$ can be derived from the prescribed strengths $r$ and the equation $\dot{u}^T f = 1$ can be written.

— Secondly, we obtain the compatible collapse displacement rates $\dot{U}_i$ from the deformations of each mode, as shown in equation (4.11).

— We adopt normalized unitary values for the $i$-th component of the $i$-th $\Lambda_i$ independent global collapse mode, with zeros in all the other components; i.e. $\Lambda_1^T = [1, 0, 0]$, $\Lambda_2^T = [0, 1, 0]$, $\Lambda_3^T = [0, 0, 1]$. This results in unitary normalized components in all resistances, $R$, and consequently, produces three columns for the global flow $\Psi^T$ matrix, each of which has a corresponding $\dot{U}_i$ vector as a result of equation (2.18). This implies a definite form for equation (2.17) which describes the load limit.

— We obtain the $L$ matrix, equation (2.19), that has as many columns as collapse modes and where each of its columns is constructed using the $\lambda$ values for the corresponding collapse mode. These $\lambda$ values are obtained while adhering to the following set of conditions:

  — The five equations that derive from each non-null deformation and the flow rule, equation (2.13) where $\psi$ and $\dot{u}$ are already given.
  — The 13 conditions required to assert the null $\lambda$ values that correspond to each selected collapse mode. This is because only five of the conditions included in the whole set—represented by the $\lambda$ multipliers, see equation (2.12)—can be considered active. The set contains conditions where one half represents limits for the positive internal forces and the other half for the negative internal forces.

Therefore, the new set amounts to 18 equations for the 18 components of $\boldsymbol{\lambda}$ for each collapse mode that is being considered.

In our example, the results are

$$\Psi = \begin{bmatrix} 0.002285 & 9 \times 10^{-5} & -2 \times 10^{-6} & -2 \times 10^{-6} & -2 \times 10^{-6} \\ 0.000204 & 0.005095 & -0.000102 & 0.000102 & 0.000102 \\ 0.001528 & 0.00191 & -3.8 \times 10^{-5} & 3.6 \times 10^{-5} & 3.6 \times 10^{-5} \end{bmatrix} \quad R^T = \begin{bmatrix} 1 & 1 & 1 \end{bmatrix}$$

and

$$L = \begin{bmatrix} 0 & 0 & 0 \\ 0 & 0 & 0 \\ 0 & 0.000203 & 0.000074 \\ 0 & 0 & 0 \\ 0 & 0 & 0 \\ 0 & 0 & 0 \\ 0 & 0 & 0 \\ 0 & 0 & 0 \\ 0.000060 & 0 & 0.000038 \\ 0 & 0 & 0 \\ 0.000058 & -0.000102 & 0 \\ 0 & 0 & 0 \\ 0.000060 & 0.000011 & 0.000042 \\ 0 & 0 & 0 \\ 0.000058 & 0.000112 & 0.000078 \\ 0 & 0 & 0 \\ 0.002062 & 0.000374 & 0.001448 \\ 0 & 0 & 0 \end{bmatrix}.$$

These results allow the reader to easily verify equation (2.23) and the limit loads previously calculated in the analysis problem.

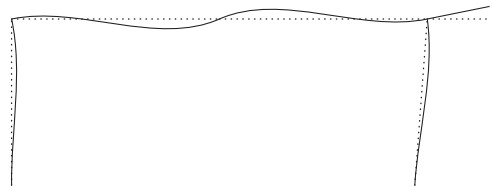

**Figure 25.** The eigen displacement mode of the stiffness matrix, no. 1.

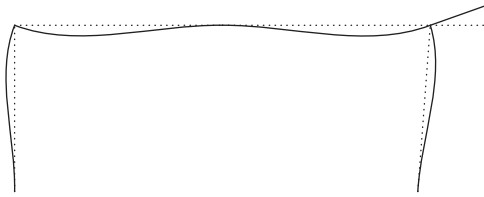

**Figure 26.** The eigen displacement mode of the stiffness matrix, no. 2.

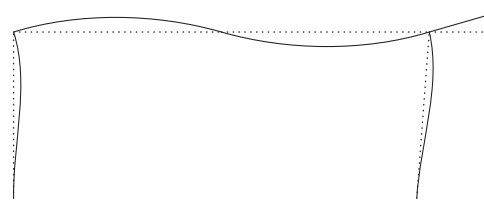

**Figure 27.** The eigen displacement mode of the stiffness matrix, no. 3.

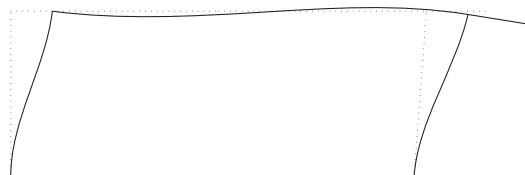

**Figure 28.** The eigen displacement mode of the stiffness matrix, no. 4.

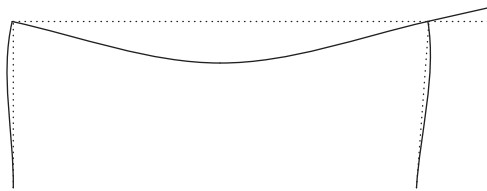

**Figure 29.** The eigen displacement mode of the stiffness matrix, no. 5.

## 5.9. Diagonalization of the stiffness matrix

The diagonalization of the elastic stiffness matrix produces the modal decomposition of loads and displacements fields ordered from the stiffest to the most flexible as shown in figures 25–29. As described in §4.4, the $F$ load and $U$ displacement vectors are strictly proportional in each of these modes.

These figures show that, according to the analysis and design, the most relevant modes are those that are the most flexible.

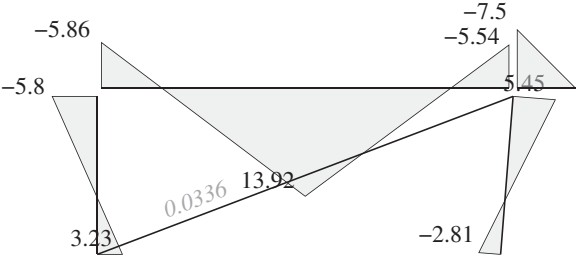

**Figure 30.** Bending moments and diagonal force for the approximated flexibility matrix with 1 mode.

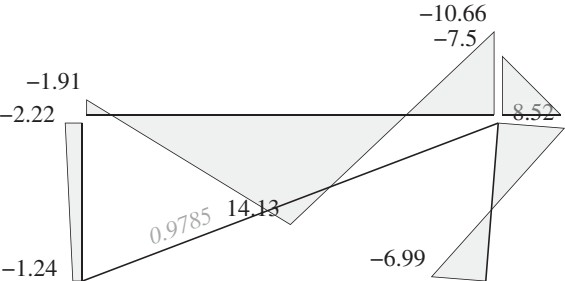

**Figure 31.** Bending moments and diagonal force for the approximated flexibility matrix with 2 modes.

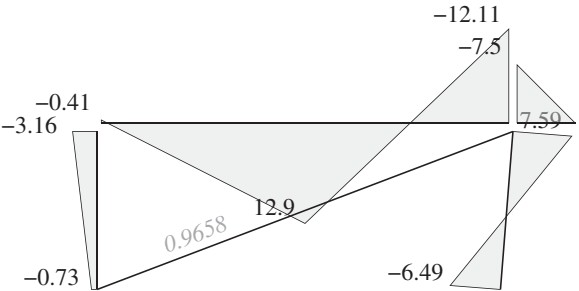

**Figure 32.** Bending moments and diagonal force for the approximated flexibility matrix with 3 modes.

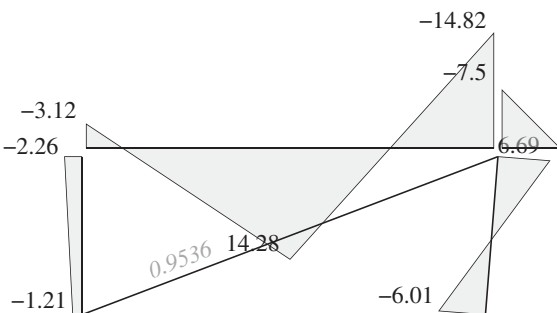

**Figure 33.** Bending moments and diagonal force for the approximated flexibility matrix with 4 modes.

To illustrate this assertion, figures 30–34 show the elastic solutions for the frame computed with an approximate flexibility matrix $K_{\approx}^{-1}$ constructed with the most flexible components, i.e. $K^{-1} \approx K_{\approx}^{-1}$, where $K_{\approx}^{-1} = \Phi_* \Omega_*^{-1} \Phi_*^T$, and $\Phi_*$ and $\Omega_*^{-1}$ are the eigenvectors of the selected modes and the diagonal matrix of the inverses of the corresponding eigenvalues in $K$, respectively. These figures demonstrate the influence of successively added modes.

Table 4 also illustrates the effective load vector $F_{\approx}$ as equilibrated by the internal forces $f_{\approx}$ for each approximation.

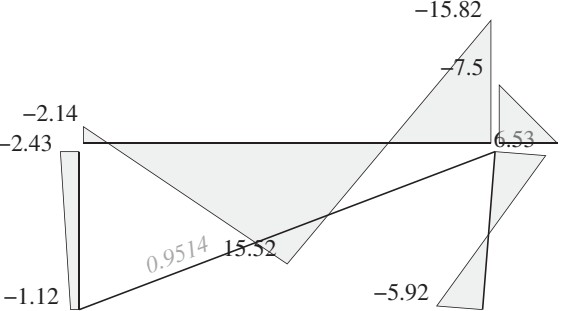

**Figure 34.** Bending moments and diagonal force for the exact flexibility matrix with 5 modes.

**Table 4.** Force vectors equilibrated by internal forces depending on the modes selected for the approximation

|            | $F_h$  | $F_v$ | $M_1$  | $M_2$  | $M_3$  |
|------------|--------|-------|--------|--------|--------|
| **5**      | −0.02  | 0.80  | −0.02  | −0.00  | 0.02   |
| **4, 5**   | 1.23   | 0.83  | −0.03  | 0.01   | −0.00  |
| **3, 4, 5**| 1.14   | 0.83  | −2.55  | 2.46   | −2.48  |
| **2, 3, 4, 5** | 1.12 | 1.00 | 1.21  | 2.49   | −6.28  |
| **all 5**  | 1.10   | 1.00  | 0.00   | −0.00  | −7.50  |

# 6. Conclusion

This paper has shown and emphasized the importance of several algebraic properties that can be found in the linear relationships between structural variables:

— We have provided a unified formulation for both elastic and limit theories, in which the only difference is in the representation of the material properties.
— This unified formulation has proven useful in the application of both theories to two different classes of problems: the analysis and the design problems. The former is devoted to the assessment of the behaviour of a known structure, while the latter is devoted to the development of a structural solution that is consistent with a predetermined behaviour.
— The analysis of the displacements generated by prestress in an elastic structure, formulated as a forced compatibility problem, has been proven to be a dual of the classic elastic problem of finding the internal forces generated by external loads.
— The unified formulation has been capable of relating the two cases studied—the elastic pre-stressing problem and the plastic ductility requirements problem—and provides a practical method of solving the latter.
— The analysis of the algebraic properties of equilibrium and compatibility matrices has facilitated both the comprehension and detection of self-stress states in hyperstatic structures or substructures and the mechanisms in hypostatic structures, thus providing a means of classifying such structures or substructures.
— The possibility of performing a dimensionality reduction of the problems via the least energetic modes, which emerge from eigenvalue or SVD problems has been shown.
— Thus, we stress the importance and usefulness of a profound understanding of the mathematical properties of structural models.

Data accessibility. All data required for the construction of the model and results used during the study can be fully recreated with the R code contained in the 'estructAlgebra.R' file supplied as electronic supplementary material in this submission.

Authors' contributions. J.C.B. provided the main ideas and the initial writing, L.N.-S. developed the examples and reviewed the manuscript.

Competing interests. The authors have no competing financial interest.

Funding. The authors receive their salary from Universidad Politécnica de Madrid and have not received additional funding for this study.

Acknowledgements. The writing of this paper has been greatly facilitated by the existence and integration of such useful tools as LaTeX (https://www.latex-project.org) for typesetting and R [20] for computation, with their huge user communities and libraries such as the CTAN and CRAN archives (https://ctan.org https://cran.r-project.org), 'knitr' [30] for integration of the aforementioned tools, and TikZ [31] for the inclusion or generation of drawings in TeX documents. We thank to Diego Cervera for his careful reading and helpful comments and recommendations, and also thank Alfonso Casal for his reading and comments. We want to thank Trevor Cattle for his collaboration in the linguistic revision of the text. Finally, we are also very grateful to the reviewers of the paper for their effort and the valuable suggestions provided for the improvement of the text.

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
