## [Peer Review File · Royal Society Open Science]

Review History

RSOS-210459.R0 (Original submission)

Review form: Reviewer 1

Is the manuscript scientifically sound in its present form?

No

Are the interpretations and conclusions justified by the results?

Yes

Is the language acceptable?

No

Do you have any ethical concerns with this paper?

No

Have you any concerns about statistical analyses in this paper?

No

Recommendation?

Major revision is needed (please make suggestions in comments)

Comments to the Author(s)

Text is difficult to follow. Numerous incorrectly used words, complex long sentences make this article virtually impossible to understand. Below are selected items but authors should revise the entire text to correct all language issues, simplify long sentences and paragraphs.

General:

- In places it seems word "symmetry" is used in place in "duality" or it does have different meaning than actual symmetry. Please confirm the correct meaning used especially on page 11 of 101 line 37 and onward.

- Numerous incorrect use of words in quotations. i.e. see "reduces" page 15 line 48, "prestress or self-tension state" page 15 line 57, "vector" page 16 line 6, "hyperstatic" page 16 line 9, "arbitrary" page 16 line 38, also see others

page 5 of 101

lines 29, 30: free movements and states of self-stress should be noted to be linearly independent

line 39: correct: "selected in order establish" to a correct form

page 6 of 101

line 13: add "... there are following restrictions:"

lines 11 to 15: reword paragraph

line 42: note is not clear

page 7 of 101:

lines 34 to 40: reword paragraph

page 11 of 101

line 46: word "also" may be incorrectly used here.

page 12 of 101

FYI: eq 2.36 is also known as "load path"

line 51: FYI: this is assuming location of loads and supports does not change during optimization

page 13 of 101

line 33: What to do with cases of multiple states of self stress where unstressed condition does not exist

line 52: to those really stressing

line 54 to 59: revise paragraph

page 14 of 101

line 11: one too many "are"

lines 39 to 49: revise paragraph

line 50: "It can be stressed..." wrong choice of word "stressed"

line 55: "proven" not "proved"

page 15 of 101

lines 47 of 52: revise paragraph

page 24 of 101: load pattern is known as load case or load combination. load pattern has different common meaning than intended

Review form: Reviewer 2

Is the manuscript scientifically sound in its present form?

No

Are the interpretations and conclusions justified by the results?

No

Is the language acceptable?

No

Do you have any ethical concerns with this paper?

No

Have you any concerns about statistical analyses in this paper?

No

Recommendation?

Major revision is needed (please make suggestions in comments)

Comments to the Author(s)

See attached file (Appendix A).

Decision letter (RSOS-210459.R0)

Dear Dr Cervera Bravo,

The Editors assigned to your paper RSOS-210459 "Prestress behaviour and ductility requirements in structures: solutions from a unified algebraic approach" have now received comments from reviewers and would like you to revise the paper in accordance with the reviewer comments and any comments from the Editors. Please note this decision does not guarantee eventual acceptance.

Please submit your revised manuscript and required files (see below) no later than 21 days from today's (ie 05-Jul-2021) date. Note: the ScholarOne system will 'lock' if submission of the revision is attempted 21 or more days after the deadline. If you do not think you will be able to meet this deadline please contact the editorial office immediately.

Please note article processing charges apply to papers accepted for publication in Royal Society Open Science (<https://royalsocietypublishing.org/rsos/charges>). Charges will also apply to papers transferred to the journal from other Royal Society Publishing journals, as well as papers submitted as part of our collaboration with the Royal Society of Chemistry

(<https://royalsocietypublishing.org/rsos/chemistry>). Fee waivers are available but must be requested when you submit your revision (<https://royalsocietypublishing.org/rsos/waivers>).

on behalf of Professor R. Kerry Rowe (Subject Editor)
openscience@royalsociety.org

Associate Editor Comments to Author:

Thank you for your submission. We have received two reports which both recommended some major revisions to your paper. Please pay particularly close attention to improving the quality of the language and grammar throughout your paper. At present, both referees note that there are sections of your manuscript which are very difficult to read (and I note many typos and grammatical errors throughout). A rewrite of these sections may take time, so please let us know if you require an extension. Please ensure to provide a tracked changes version of your paper (as well as a 'clean' version), and a thorough point-by-point response to the reviewers. Best wishes

Reviewer comments to Author:

Reviewer: 1

Comments to the Author(s)

Text is difficult to follow. Numerous incorrectly used words, complex long sentences make this article virtually impossible to understand. Below are selected items but authors should revise the entire text to correct all language issues, simplify long sentences and paragraphs.

General:

- In places it seems word "symmetry" is used in place in "duality" or it does have different meaning than actual symmetry. Please confirm the correct meaning used especially on page 11 of 101 line 37 and onward.

- Numerous incorrect use of words in quotations. i.e. see "reduces" page 15 line 48, "prestress or self-tension state" page 15 line 57, "vector" page 16 line 6, "hyperstatic" page 16 line 9, "arbitrary" page 16 line 38, also see others

page 5 of 101

lines 29, 30: free movements and states of self-stress should be noted to be linearly independent

line 39: correct: "selected in order stablish" to a correct form

page 6 of 101

line 13: add "... there are following restrictions:"

lines 11 to 15: reword paragraph

line 42: note is not clear

page 7 of 101:

lines 34 to 40: reword paragraph

page 11 of 101

line 46: word "also" may be incorrectly used here.

page 12 of 101

FYI: eq 2.36 is also known as "load path"

line 51: FYI: this is assuming location of loads and supports does not change during optimization

page 13 of 101

line 33: What to do with cases of multiple states of self stress where unstressed condition does not exist

line 52: to those really stressing

line 54 to 59: revise paragraph

page 14 of 101

line 11: one too many "are"

lines 39 to 49: revise paragraph

line 50: "It can be stressed..." wrong choice of word "stressed"

line 55: "proven" not "proved"

page 15 of 101

lines 47 of 52: revise paragraph

page 24 of 101: load pattern is known as load case or load combination. load pattern has different common meaning than intended

Reviewer: 2

Comments to the Author(s)

see attached file

===PREPARING YOUR MANUSCRIPT===

===PREPARING YOUR REVISION IN SCHOLARONE===

Author's Response to Decision Letter for (RSOS-210459.R0)

See Appendix B.

RSOS-210459.R1 (Revision)

Review form: Reviewer 1

Is the manuscript scientifically sound in its present form?

Yes

Are the interpretations and conclusions justified by the results?

Yes

Is the language acceptable?

Yes

Do you have any ethical concerns with this paper?

No

Have you any concerns about statistical analyses in this paper?

No

Recommendation?

Accept with minor revision (please list in comments)

Comments to the Author(s)

I do have a repeated comment on quotations. I do understand the quotations are used to emphasize a word. However, do these words need to be emphasized? If so I propose to use bold font rather than the quotations. This is because quotations have ambiguous meaning. If I say: I had "fresh" ham in my sandwich. It would not mean the ham was especially fresh but rather questionable fresh, wouldn't it?

Review form: Reviewer 2

Is the manuscript scientifically sound in its present form?

Yes

Are the interpretations and conclusions justified by the results?

Yes

Is the language acceptable?

Yes

Do you have any ethical concerns with this paper?

No

Have you any concerns about statistical analyses in this paper?

No

Recommendation?

Accept with minor revision (please list in comments)

Comments to the Author(s)

See attached report (Appendix C).

Decision letter (RSOS-210459.R1)

Dear Dr CERVERA BRAVO

On behalf of the Editors, we are pleased to inform you that your Manuscript RSOS-210459.R1 "Prestress behaviour and ductility requirements in structures: solutions from a unified algebraic approach" has been accepted for publication in Royal Society Open Science subject to minor revision in accordance with the referees' reports. Please find the referees' comments along with any feedback from the Editors below my signature.

Please submit your revised manuscript and required files (see below) no later than 7 days from today's (ie 11-Oct-2021) date. Note: the ScholarOne system will 'lock' if submission of the revision is attempted 7 or more days after the deadline. If you do not think you will be able to meet this deadline please contact the editorial office immediately.

Kind regards,

on behalf of R. Kerry Rowe (Subject Editor)
openscience@royalsociety.org

Associate Editor Comments to Author:

Comments to the Author:

Thank you for tackling the reviewers' earlier concerns - it seems that your efforts have improved the paper; however, there remain a number of questions and matters that need to be addressed. These do not appear to preclude the possibility of publishing the paper, but you must provide responses to the concerns raised in this round of review and also demonstrate to the editors that you are not only responding but incorporating the changes that are most pertinent: Reviewer 2, for instance, raises a number of questions regarding the literature review conducted. Please make sure you take these questions into consideration - they would add value to your work in the editor's view.

Reviewer comments to Author:

Reviewer: 1

Comments to the Author(s)

I do have a repeated comment on quotations. I do understand the quotations are used to emphasize a word. However, do these words need to be emphasized? If so I propose to use bold font rather than the quotations. This is because quotations have ambiguous meaning. If I say: I had "fresh" ham in my sandwich. It would not mean the ham was especially fresh but rather questionable fresh, wouldn't it?

Reviewer: 2

Comments to the Author(s)

See attached report.

===PREPARING YOUR MANUSCRIPT===

===PREPARING YOUR REVISION IN SCHOLARONE===

Author's Response to Decision Letter for (RSOS-210459.R1)

See Appendix D.

Decision letter (RSOS-210459.R2)

Dear Dr Cervera Bravo,

I am pleased to inform you that your manuscript entitled "Prestress behaviour and ductility requirements in structures: solutions from a unified algebraic approach" is now accepted for publication in Royal Society Open Science.

on behalf of Professor R. Kerry Rowe (Subject Editor)
openscience@royalsociety.org

Appendix A

Review of

Prestress behaviour and ductility requirements in structures: solutions for a unified algebraic approach by Jaime Cervera Bravo and Laura Navas-Sanchez

This paper claims to make some rather profound contributions to the theory of structures. For example, the second sentence of the abstract claims that “a previously unnoticed duality between two different problems is shown: the elastic solution to the internal forces of an externally loaded structure and the nodal displacements induced by prestress in one or several elements of this same structure”. However, the paper is so badly written that I could not find where this new duality was described in the paper.

The quality of writing is very low, making it very difficult to read and follow. Much of the early material is well-known, and yet the brief reference list fails to cite many of the key papers on this very subject, by Kuznetsov, Argyris, Calladine or Pellegrino, say, to say nothing of earlier pioneers such as Castigliano or Muller-Breslau.

An example is given in Section 5 which is not very instructive, and only serves to confuse. If a new duality has indeed been uncovered somewhere in all this confusion then surely a much simpler example would be available with which to demonstrate it and make it clear.

I followed the example in some detail. It did not seem to be incorrect. It seemed to be essentially some fairly standard linear structural analysis, but with a very unusual choice of basis vectors. Quite why those basis vectors were chosen remains a mystery to me – I could not see any explanation, nor could I see whether there was any advantage to be gained by taking this rather circuitous route.

In summary, I do not consider that the paper is suitable for publication. It would need substantial revisions to clarify what the contributions are and why they are important. Given that at this stage I am unable to discern what the contributions are, I cannot say whether they are worthy of publication.

Appendix B

Response to reviewers and editor's questions

Ordered as the guidelines in the "PREPARING YOUR MANUSCRIPT" and following sections of the "Royal Society Open Science - Decision on Manuscript ID RSOS-210459" e-mail,

===PREPARING OUR MANUSCRIPT===

1: The files of both versions included are

- one version identifying all the changes that have been made: `estructAlgebraTracked.pdf`
- a 'clean' version of the new manuscript: `estructAlgebra.pdf`
 - As said below, both pdf files can be generated with the same LaTeX files included in the LaTeX zip folder changing a macro in the LaTeX main file: `estructAlgebra.tex`.

2: All equations are editable as the manuscript is in LaTeX format.

3: The acknowledgements section is included before reference list/bibliography

4: The bibliography is in Vancouver style as a result of using the latex class "rsos.cls" and the RSOS bibliography style "RS.bst".

5: In order to improve the language and flow of the text we have employed the assistance of professional services. For that reason, we have mentioned Trevor Cattle (Top English Academy) in the acknowledgements section, and added a professional language editing service from Cambridge Proofreading LLC. We enclose this certificate in the file: `certificate_9753190.pdf` that has been classified in the Manuscript Central category of "Recommendation file"

===PREPARING YOUR REVISION IN SCHOLARONE===

0: Revision has been created in <https://mc.manuscriptcentral.com/rsos> the 9th of August, and after the return to the draft state the 19th of August, reorganized and resubmitted the 20th of August

1: The point-by-point response to referees and editors requested in Step 1 'View and respond to decision letter' is provided herein (file: "Answers_to_referees_and_editors.docx"). Responses to referees are in the following sections of this document:

===RESPONSES TO REFEREE 1===, ===RESPONSES TO REFEREE 2===

2: The summary requested in Step 2 "Type, Title, & Abstract" of no more of 100 words addressed to a non-scientific audience is also included herein:

Type: Research article

Title: Prestress behaviour and ductility requirements in structures: solutions from a unified algebraic approach

Abstract:

Mathematics are crucial to comprehend the behaviour of structures. For that reason, this article presents an algebraic approach that unifies both the elastic and limit theories of static structural analysis. This approach reveals

a novel mathematical relationship between structural problems that permits the calculation of the ductility requirements necessary to achieve full plastic behaviour; among other contributions. Moreover, the dual relationship between hyperstatic and hypostatic structural classes of problems has been thoroughly discussed. Finally, we present the dimensionality reduction of problems via the application of a number of well-known mathematical tools and illustrate it with an example.

3: The uploaded files are

- 1st document: (tracked version), estructureAlgebraTracked... in pdf format.
- 2nd document: (clean version) estructureAlgebra... in pdf format.
 - TeX file can generate both pdfs, by changing a macro in LaTeX, that activates a different behavior in the compilation, in lines 86 and 88 of the main LaTeX file.: estructureAlgebra.tex included in the Latex zip folder.
- All the files required in the latex framework for the compilation: style and bibliography files, and figures and captions in latex and pdf format. I have collected all LaTeX files required for compilation in a unique zip folder: LatexFiles_RSOS-210459.zip. It should be noted:
 - that there are figures that are directly inlined in the LaTeX code and that can be perfectly seen (and reviewed) in the pdf file, in the "clean" version.
 - that all captions are included and can be edited in the LaTeX code
 - that tables are formatted and can be edited in the LaTeX code
 - that the lists of the tables and the figures can be generated with the LaTeX macros included at the end of the document in the estructureAlgebra.tex file: macros \listoftables, \listoffigures. Anyhow, we include the files ListOfTables.docx and ListOfFigures.docx for reference in the (redundant) zip files, the first with each of the figures generated with LaTeX: FiguresDuplicated_RSOS-210459.zip, the second TableReferences_RSOS-210459.zip
- The supplementary material (R code) : file estructureAlgebra.R
- A copy of this file (our point-by-point response to referees and Editors)
- The certificate of the language editing support: certificate_9753190.pdf

4: Step 6:

Queries on the electronic submission form:

- Data-access statement: all data can be recreated with the R code supplied in the ESM
- ESM
 - file: estructureAlgebra.R,
 - Title: R code to recreate the example in Section 5 of "Prestress behaviour and ductility requirements in structures: solutions from a unified algebraic approach"
 - Caption: This file contains the R code that has been used to the construction of the model of the Frame, to solve all the problems, and to draw all the figures of these solutions, as presented in Section 5. The file was extracted from the original *.Rnw file that contains both the LaTeX and the R code that generates the entire paper.

5: Step 7: 'Review & submit' .

===RESPONSES TO REFEREE 1===

Text is difficult to follow. Numerous incorrectly used words, complex long sentences make this article virtually impossible to understand. Below are selected items but authors should revise the entire text to correct all language issues, simplify long sentences and paragraphs.

We have endeavoured to improve the readability, comprehension and fluidity of the text. For that reason, we have revised it completely with the assistance of an English teacher and submitted our work to a professional language editing service.

General:

- In places it seems word "symmetry" is used in place in "duality" or it does have different meaning than actual symmetry. Please confirm the correct meaning used especially on page 11 of 101 line 37 and onward.

We have eliminated the word *symmetry* and substituted it with the word *duality* throughout the text, particularly in subsection 2.3.

- Numerous incorrect use of words in quotations. i.e. see "reduces" page 15 line 48, "prestress or self-tension state" page 15 line 57, "vector" page 16 line 6, "hyperstatic" page 16 line 9, "arbitrary" page 16 line 38, also see others

The main purpose of most quotations was to emphasize and contextualize the importance of the concept involved. In any case, we have removed a number of them.

"reduces" page 15 line 48:

We used the quotation marks as the reduction is complete and eliminates the whole gap.

"prestress or self-tension state" page 15 line 57:

In the new version of the manuscript, we have used the term "self-stress" throughout the text. See in particular change No. 494.

The quotations were there to emphasize the relevance of such state, but have been removed.

"vector" page 16 line 6, "hyperstatic" page 16 line 9, "arbitrary" page 16 line 38, also see others

The quotations have been removed.

page 5 of 101

lines 29, 30: free movements and states of self-stress should be noted to be linearly independent

This very pertinent remark has been included in our work, see change No. 84.

line 39: correct: "selected in order stablish" to a correct form

This sentence has been corrected: selected in order to establish, change No. 87.

page 6 of 101

line 13: add "... there are following restrictions:" and lines 11 to 15: reword paragraph.

We have revised the paragraph in order to make it clearer, changes No. 101, ff.

line 42: note is not clear

The note shows the algebraic equivalences to the symbols adopted in the schema.

We have added a sentence that clarifies it.

page 7 of 101:

lines 34 to 40: reword paragraph

We have revised the paragraph in order to make it clearer, changes No.140 ff.

page 11 of 101

line 46: word "also" may be incorrectly used here.

The whole sentence has been rephrased, change No. 298

page 12 of 101

FYI: eq 2.36 is also known as "load path"

We have included your comment in the text, change No. 330

line 51: FYI: this is assuming location of loads and supports does not change during optimization

We have included your comment in the text, change No. 345.

page 13 of 101

line 33: What to do with cases of multiple states of self stress where unstressed condition does not exist

We have revised the paragraph in order to make it clear that this unstressed condition corresponds to the isolated and unconnected members, change No. 372 ff.

line 52: to those really stressing line 54 to 59: revise paragraph

We have revised the text in order to make it clearer and more correct, changes No. 389, ff.

page 14 of 101

line 11: one too many "are"

This error has been corrected, change No. 401.

lines 39 to 49: revise paragraph

We have revised the paragraph in order to make it clearer, changes No. 421, ff.

line 50: "It can be stressed..." wrong choice of word "stressed"

This expression has been rectified, change No. 439.

line 55: "proven" not "proved"

This error has been rectified throughout the text

page 15 of 101

lines 47 of 52: revise paragraph

We have revised the paragraph in order to make it clearer. Changes 486, ff.

page 24 of 101: load pattern is known as load case or load combination. load pattern has different common meaning than intended

The expression has been rectified changes No. 809, and others...

===RESPONSES TO REFEREE 2===

Prestress behaviour and ductility requirements in structures: solutions for a unified algebraic approach by Jaime Cervera Bravo and Laura Navas-Sanchez

This paper claims to make some rather profound contributions to the theory of structures. For example, the second sentence of the abstract claims that “a previously unnoticed duality between two different problems is shown: the elastic solution to the internal forces of an externally loaded structure and the nodal displacements induced by prestress in one or several elements of this same structure”. However, the paper is so badly written that I could not find where this new duality was described in the paper.

The text has been completely revised with an English teacher and submitted to a professional language editing service in order to improve its readability, comprehension and fluidity.

The previously unnoticed duality between two different problems mentioned is now emphasized in section 3 b.

The quality of writing is very low, making it very difficult to read and follow.

We have endeavoured to improve the quality of the writing.

Much of the early material is well-known, and yet the brief reference list fails to cite many of the key papers on this very subject, by Kuznetsov, Argyris, Calladine or Pellegrino, say, to say nothing of earlier pioneers such as Castigliano or Muller-Breslau.

As the article does not pretend to be a historical review, we had adhered to classic texts that provide a comprehensive framework to such theories. Still we have added a key reference to Argyris and a reference to an early review of the issue, by Clough, see change No. 25.

An example is given in Section 5 which is not very instructive, and only serves to confuse. If a new duality has indeed been uncovered somewhere in all this confusion then surely a much simpler example would be available with which to demonstrate it and make it clear.

We have not changed the example because we consider that a simpler example would not permit us to integrate all the issues discussed throughout the paper. Furthermore, a simpler example could transmit the idea that the scope of the presented mathematical relationships and methods, including the new duality or the procedure to compute the ductility requirements, are limited to examples with a reduced number of joints or loads, and this is not true. We want to show that the proposed procedures can be applied to an actual structure.

I followed the example in some detail. It did not seem to be incorrect. It seemed to be essentially some fairly standard linear structural analysis, but with a very unusual choice of basis vectors. Quite why those basis vectors were chosen remains a mystery to me – I could not see any explanation, nor could I see whether there was any advantage to be gained by taking this rather circuitous route.

This is correct: this is a linear elastic and limit analysis. The purpose of the example is to show all procedures and concepts in a non-trivial structure but also in one where every detail can be accurately tracked. Thus we have proposed a reduced model in which the possibility of unifying the elastic and plastic approaches can be verified and to which all the mathematical models presented in the article can be applied.

In summary, I do not consider that the paper is suitable for publication. It would need substantial revisions to clarify what the contributions are and why they are important. Given that at this stage I am unable to discern what the contributions are, I cannot say whether they are worthy of publication.

We hope that the revisions made and the improvement in the quality of the text allow you to value positively our paper, and the contributions that are included therein.

Appendix C

Review of Revised version of “Prestress behaviour etc” by Cervera Bravo and Navas-Sanchez

The revised document still has many shortcomings, but it is a substantial improvement on the one that was originally submitted. The language and grammar are much improved, and the professional proof-reading was no doubt a major factor in that. (A number of typographical and grammatical errors still remain, however). The logic of the paper is now much easier to follow, and the explicit statement in Section 3(b) of exactly what constitutes this purported new duality was a very helpful navigational aid for the reader.

I was rather surprised at the response to my request for a proper review of the literature. A couple of references were added, and they were referred to rather briefly and brusquely. The authors’ response that the paper was not “a historical review” was rather clumsy.

When asked for more references to the pre-existing knowledge in this area, the choice of the Clough paper is a rather unusual and disappointing response. The paper contains no equations (other than a very general, rather trivial and here utterly irrelevant trio related to axisymmetry). It is largely an anecdotal account of the personalities involved in the development of finite element theory. Here is a representative passage from that paper:

During the academic year 1964-65 Clough was a Visiting Professor at Cambridge University. He continued to supervise a number of students by mail, met other international experts in the field and wrote several research papers on earthquake engineering and the finite element method. During this period Dr. Zienkiewicz, then installed as a Professor at the University of Wales in Swansea, asked Clough and many other leading specialists on the development of new methods of analysis to take part in a conference on Stress Analysis at the University.

That sample, and the rest of that paper, does not in any way engage intellectually with the issues of mechanics at hand in the paper submitted here. I am astonished that the authors think that adding one reference to Argyris and this utterly unhelpful reference to Clough in any way responds to my request for a proper synopsis of the previous work in this area. A proper statement of the existing technical literature in this area would then allow reviewers and readers to see in what sense the current paper makes an original contribution, if any.

The reference to the Maxwell 1870 paper is unusual. The authors state “However, other authors propose the use of the “self-strain” approach, which are more difficult to interpret [5]”. I am not aware of anything in the 1870 paper that could be described as a “self-strain approach”. Moreover, on what basis can the authors simply state that Maxwell’s approach – if indeed there is such - is “more difficult to interpret”? The authors’ sentence begs very many more questions than it answers.

It took me a while to work out what Equations f1-f4 were trying to say i.e. that they were just trying to explain the symbols used in Eqn 2.7, for example. So for example “TAG” might mean either “COM” or “EQU”. It was all a little elaborate, and I am not sure whether it made matters clearer or more obscure.

I still believe the paper is badly written on many levels, but fundamentally I do not think its propositions are incorrect. I therefore recommend that it be accepted for publication, very much in the spirit of Open Science, such that the wider community can look at this work and make their own judgements as to its importance, without the wider debate being held back by the refereeing process. The paper satisfies the criteria in the Guidelines for Referees document – it is original and it advances scientific knowledge, etc.

Minor typographical errors. (List incomplete)

Page 1 Line 37 needed/needed

Page 1 Line 40 “allows to obtain the”/ “allows the to be obtained.”

Page 2 line 18 mew/new

Page 2 line 44 analized/analysed

Page 4 line 4 “linear applications” – should that be “linear maps”? I do not know what a linear application is.

There were more.

Appendix D

Cover Letter including the Response to the comments and questions from the Associate Editor and the reviewers:

Questions and comments are in red followed by responses in black.

We add also detailed explanations of the resubmission steps, ordered as the guidelines in the "PREPARING YOUR MANUSCRIPT" and following sections of the e-mail "Royal Society Open Science - Decision on Manuscript ID RSOS-210459.R1"

Associate Editor Comments to Author:

Thank you for tackling the reviewers' earlier concerns - it seems that your efforts have improved the paper; however, there remain a number of questions and matters that need to be addressed. These do not appear to preclude the possibility of publishing the paper, but you must provide responses to the concerns raised in this round of review and also demonstrate to the editors that you are not only responding but incorporating the changes that are most pertinent: Reviewer 2, for instance, raises a number of questions regarding the literature review conducted. Please make sure you take these questions into consideration - they would add value to your work in the editor's view.

We do acknowledge the valuable reviewers' comments. For that reason, we have endeavoured to implement every issue mentioned in order to improve the text. Furthermore, we have decided to mention the reviewers in the article as a symbol of our gratitude for their efforts and suggestions.

===REVIEWER 1===

Comments to the Author(s)

I do have a repeated comment on quotations. I do understand the quotations are used to emphasize a word. However, do these words need to be emphasized? If so I propose to use bold font rather than the quotations. This is because quotations have ambiguous meaning. If I say: I had "fresh" ham in my sandwich. It would not mean the ham was especially fresh but rather questionable fresh, wouldn't it?

We agree with this appreciation, thus in a general way we have decided to eliminate the quotation marks used to emphasize the words. Just in the cases we have considered the term should be emphasised, we have made use of the Latex macro $\emph{\{}}$. And uniquely in two particular cases we have maintained the quotations marks. The former is the first occurrence of the variable u , which represents deformations or relative internal movements of the nodes of bars or elements. And the second is the term the "*horizontal*" forces mentioned in the section that includes the example. This is because this term refers to the horizontal forces involved as a consequence of the vertical forces in the slanted column.

Corrections (5, 7, 8, 22, 23, 24, 28 to 40, 42, etc, ...)

===REVIEWER 2===

The revised document still has many shortcomings, but it is a substantial improvement on the one that was originally submitted. The language and grammar are much improved, and the professional proof-reading was no doubt a major factor in that. (A number of typographical and grammatical errors still remain, however). The logic of the paper is now much easier to follow, and the explicit statement in Section 3(b) of exactly what constitutes this purported new duality was a very helpful navigational aid for the reader.

We are also very grateful to the Reviewer 2 for the valuable suggestions provided towards the improvement of the document, as well as for his gratifying recognition of our effort to improve our document.

I was rather surprised at the response to my request for a proper review of the literature. A couple of references were added, and they were referred to rather briefly and brusquely. The authors' response that the paper was not "a historical review" was rather clumsy.

When asked for more references to the pre-existing knowledge in this area, the choice of the Clough paper is a rather unusual and disappointing response. The paper contains no equations (other than a very general, rather trivial and here utterly irrelevant trio related to axisymmetry). It is largely an anecdotal account of the personalities involved in the development of finite element theory. Here is a representative passage from that paper:

" During the academic year [...] a conference on Stress Analysis at the University."

That sample, and the rest of that paper, does not in any way engage intellectually with the issues of mechanics at hand in the paper submitted here. I am astonished that the authors think that adding one reference to Argyris and this utterly unhelpful reference to Clough in any way responds to my request for a proper synopsis of the previous work in this area. A proper statement of the existing technical literature in this area would then allow reviewers and readers to see in what sense the current paper makes an original contribution, if any.

The reference to Clough 1990 was intended to indicate part of the relevant literature in an indirect way. In our opinion, the interpretation of the evolution of knowledge and of the methods in structural analysis, and also in algebraic and numerical methods can be considered controversial. This would be due to several factors: it involves an enormous number of authors and works (77 works and 63 authors only in the bibliography of Clough 1990), in a set of linked lines of progress, whose interpretation cannot be separated from the parallel progress in the techniques and the hardware available for computing. In this respect, see for example Kurrer 2018, chapters 11 and 12, and consider the Argyris' statement "The computer shapes the theory". The fact of explicitly quoting only the latter author was due to our conviction of the importance of the reduction of the theory to the energy methods provided by this given the excellent algebraic properties resulting from the use of the scalar product. In any case, we have included in the revised version some of the works that we believe have been key in the evolution of the matter, from its beginning perhaps in Mohr to recent compendiums, by Zienkiewicz and Atluri. Corrections (9 to 20)

The reference to the Maxwell 1870 paper is unusual. The authors state " However, other authors propose the use of the "self-strain" approach, which are more difficult to interpret [5]". I am not aware of anything in the 1870 paper that could be described as a "self-strain approach". Moreover, on what basis can the authors simply state that Maxwell's approach – if indeed there is such - is "more difficult to interpret"? The authors' sentence begs very many more questions than it answers.

The self-strain concept appears in page 12 of the paper Maxwell (1870) (Transactions, Vol XXVI part1, see <https://digital.nls.uk/scientists/archive/74629046>). There, we can read:

" If, however, in any partial system of s' points connected by e' lines, the quantity $p' = 2s' - e' - 3$ be negative, or in other words, if a part of the frame be self-strained, this partial system will contribute only $2s' - 3$ equations independent of each other to the complete system, and the whole frame will have $p - p'$ degrees of freedom. "

Obviously, this is not at a self-strain approach but rather an application of the existence of states of *self-strain* to the analysis of the indeterminacy of a structural model. Therefore, we have rewritten our text in order to avoid confusion.

The assertion about the easier understanding of the self-stress over the self-strain interpretation of the stressed state was motivated by the fact that most engineers and engineering students are more

at ease with the static than the kinematics approach in most problems, as the training in the local equilibrium is a key component in the syllabus of all structural training programs. We had no intention at all of considering Maxwell incomprehensible. On the contrary, from our point of view, he is an excellent and even topical author in many aspects.

Corrections (41 to 44)

It took me a while to work out what Equations f1-f4 were trying to say i.e. that they were just trying to explain the symbols used in Eqn 2.7, for example. So for example “TAG” might mean either “COM” or “EQU”. It was all a little elaborate, and I am not sure whether it made matters clearer or more obscure.

In the footnote in which equations f1 to f4 are included, we have added a clear reference to the schema (2.7) to point out clearly that the formulas explain the symbolism employed in the compact form adopted for such a schema. Correction (50)

I still believe the paper is badly written on many levels, but fundamentally I do not think its propositions are incorrect. I therefore recommend that it be accepted for publication, very much in the spirit of Open Science, such that the wider community can look at this work and make their own judgments as to its importance, without the wider debate being held back by the refereeing process. The paper satisfies the criteria in the Guidelines for Referees document – it is original and it advances scientific knowledge, etc.

We have tried to improve the writing of the article and have corrected all the errors explicitly pointed out in the following lines of the report:

Minor typographical errors. (List incomplete)

Page 1 Line 37 needeed/needed correction (1)

Page 1 Line 40 “allows to obtain the”/ “allows the to be obtained.” corrections (2, 3)

Page 2 line 18 mew/new correction (4)

Page 2 line 44 analized/analysed (and more...) corrections (6, 21, 61, 82)

Page 4 line 4 “linear applications” – should that be “linear maps”? I do not know what a linear application is.

Effectively, *linear application* has been an unfortunate literal translation of the term Spanish for referring to such a concept. Corrections (51, 136, 139, 141, 148)

There were more.

We have performed a final reading aiming to detect and correct potential outstanding typing errors and misleading grammatical or conceptual constructs.

Clarifications related to the guidelines in the "PREPARING YOUR MANUSCRIPT" and following sections of the "Royal Society Open Science - Decision on Manuscript ID RSOS-210459.R1" e-mail, from 2021-10-11

===PREPARING YOUR MANUSCRIPT===

1: The files of both versions included are

- one version identifying all the changes that have been made: `estructAlgebraTracked.pdf`
- a 'clean' version of the new manuscript: `estructAlgebra.pdf`
 - As said below, both pdf files can be generated with the same LaTeX files included in the LaTeX zip folder changing a macro in the LaTeX main file: `estructAlgebra.tex`.

2: All equations are editable as the manuscript is in LaTeX format.

3: The acknowledgements section is included before reference list/bibliography

4: The bibliography is in Vancouver style as a result of using the latex class "rsos.cls" and the RSOS bibliography style "RS.bst".

===PREPARING YOUR REVISION IN SCHOLARONE===

0: Revision has been created in <https://mc.manuscriptcentral.com/rsos> the 26th october and submitted the same day.

1: The point-by-point response to referees and editors requested in Step 1 'View and respond to decision letter' is provided herein (file: "Answers_to_referees_and_editors.R2.docx"). Responses to referees are in the preceding sections of this document:
"===REVIEWER 1=== ", "===REVIEWER 2=== "

2: The summary requested in Step 2 'Type, Title, & Abstract' of no more of 100 words adressed to a non-scientific audience is also included herein:

Type: Research article

Title: Prestress behaviour and ductility requirements in structures: solutions from a unified algebraic approach

Abstract:

Mathematics are crucial to comprehend the behaviour of structures. For that reason, this article presents an algebraic approach that unifies both the elastic and limit theories of static structural analysis. This approach reveals a novel mathematical relationship between structural problems that permits the calculation of the ductility requirements necessary to achieve full plastic behaviour, among other contributions. Moreover, the dual relationship between hyperstatic and hypostatic structural classes of problems has been thoroughly discussed. Finally, we present the dimensionality reduction of problems via the application of a number of well-known mathematical tools and illustrate it with an example.

3: The uploaded files are

- 1st document: (tracked version), estructureAlgebraTracked... in pdf format.
- 2nd document: (clean version) estructureAlgebra... in pdf and tex format.
 - TeX file can generate both pdfs (albeit under the same name of the original in LaTeX, name that can be renamed after the compilation in order to generate the "estructureAlgebraTracked.pdf" file), by changing a macro in LaTeX that activates a different behavior in the compilation, in lines 86 and 88 of the main LaTeX file.: estructureAlgebra.tex included in the Latex zip folder.
- All the files required in the latex framework for the compilation: style and bibliography files, and figures and captions in latex and pdf format. I have collected all LaTeX files required for compilation in a unique zip folder: LatexFiles_RSOS-210459.R2.zip. It should be noted:
 - that there are figures that are included in the LaTeX code with the `\import` macro. The imported tex files compose the figure from multiple paged pdfs, that were generated with Inkscape. Other figures are included indirectly in their numerical order via the "tikzpicture" environment, that calls the tikzexternal.sty file. This latter activates the `\includegraphics` macro. All figures can be perfectly seen (and reviewed) in the pdf file, in the "clean" version.
 - that all captions are included and can be edited in the LaTeX code
 - that tables are formatted and can be edited in the LaTeX code
 - that the lists of the tables and the figures can be generated with the LaTeX macros included at the end of the document in the estructureAlgebra.tex file: macros `\listoftables`, `\listoffigures`. Anyhow, we include the files ListOfTables.docx and ListOfFigures.docx for reference in the following (redundant) zip files, the first
 - that includes each of the figures generated with LaTeX: FiguresDuplicated_RSOS-210459.R2.zip, the second TableReferences_RSOS-210459.R2.zip
- The supplementary material (R code) : file estructureAlgebra.R
- A copy of this file (our point-by-point response to referees and Editors)

4: Step 6:

Queries on the electronic submission form:

- Data-access statement: all data can be recreated with the R code supplied in the ESM
- ESM
 - file: estructureAlgebra.R,
 - Title: R code to recreate the example in Section 5 of "Prestress behaviour and ductility requirements in structures: solutions from a unified algebraic approach"
 - Caption: This file contains the R code that has been used to the construction of the model of the Frame, to solve all the problems, and to draw all the figures of these solutions, as presented in Section 5. The file was extracted from the original *.Rnw file that contains both the LaTeX and the R code that generates the entire paper.

5: Step 7: 'Review & submit' .